# Structural characterization of ligand binding and pH-specific enzymatic activity of mouse Acidic Mammalian Chitinase

Roberto Efraín Díaz[1,2], Andrew K Ecker[3,4], Galen J Correy[1], Pooja Asthana[1], Iris D Young[1], Bryan Faust[3,5,6], Michael C Thompson[7,8], Ian B Seiple[3,4], Steven Van Dyken[9], Richard M Locksley[10,11,12], James S Fraser[1]*

[1]Department of Bioengineering and Therapeutic Sciences, University of California, San Francisco, San Francisco, United States; [2]Tetrad Graduate Program, University of California, San Francisco, San Francisco, United States; [3]Department of Pharmaceutical Chemistry, University of California, San Francisco, San Francisco, United States; [4]Cardiovascular Research Institute, University of California, San Francisco, San Francisco, United States; [5]Department of Biochemistry and Biophysics, University of California, San Francisco, San Francisco, United States; [6]Biophysics Graduate Program, University of California, San Francisco, San Francisco, United States; [7]Chemistry and Chemical Biology Graduate Program, University of California, San Francisco, San Francisco, United States; [8]Department of Chemistry and Chemical Biology, University of California, Merced, Merced, United States; [9]Department of Pathology and Immunology, Washington University School of Medicine in St Louis, St Louis, United States; [10]Department of Medicine, University of California, San Francisco, San Francisco, United States; [11]Department of Microbiology and Immunology, University of California, San Francisco, San Francisco, United States; [12]University of California, Howard Hughes Medical Institute, San Francisco, San Francisco, United States

*For correspondence:
jfraser@fraserlab.com

**Abstract** Chitin is an abundant biopolymer and pathogen-associated molecular pattern that stimulates a host innate immune response. Mammals express chitin-binding and chitin-degrading proteins to remove chitin from the body. One of these proteins, Acidic Mammalian Chitinase (AMCase), is an enzyme known for its ability to function under acidic conditions in the stomach but is also active in tissues with more neutral pHs, such as the lung. Here, we used a combination of biochemical, structural, and computational modeling approaches to examine how the mouse homolog (mAMCase) can act in both acidic and neutral environments. We measured kinetic properties of mAMCase activity across a broad pH range, quantifying its unusual dual activity optima at pH 2 and 7. We also solved high-resolution crystal structures of mAMCase in complex with oligomeric GlcNAcn, the building block of chitin, where we identified extensive conformational ligand heterogeneity. Leveraging these data, we conducted molecular dynamics simulations that suggest how a key catalytic residue could be protonated via distinct mechanisms in each of the two environmental pH ranges. These results integrate structural, biochemical, and computational approaches to deliver a more complete understanding of the catalytic mechanism governing mAMCase activity at different pH. Engineering proteins with tunable pH optima may provide new opportunities to develop improved enzyme variants, including AMCase, for therapeutic purposes in chitin degradation.

## eLife assessment

This structural and biochemical study of the mouse homolog of acidic mammalian chitinase (AMCase) enhances our understanding of the pH-dependent activity and catalytic properties of mouse AMCase, and it sheds light on its adaptation to different physiological pH environments. The methods and analysis of data are **solid**, providing several lines of evidence to support the development of mechanistic hypotheses. While the findings and interpretation will be **valuable** to those studying AMCase in mice, the broader significance, including extension of the results to other species including human, remain less clear.

## Introduction

Chitin, a polymer of β(1-4)-linked *N*-acetyl-D-glucosamine (GlcNAc), is the second most abundant polysaccharide in nature. Chitin is present in numerous pathogens, such as nematode parasites, dust mites, and fungi (*Cabib and Bowers, 1975*; *Zhu et al., 2016*; *Tang et al., 2015*), and is a pathogen-associated molecular pattern (PAMP) that activates mammalian innate immunity (*Elieh Ali Komi et al., 2018*). To mitigate constant exposure to environmental chitin, mammals have evolved unusual multi-gene loci that are highly conserved and encode chitin-response machinery, including chitin-binding (chi-lectins) and chitin-degrading (chitinases) proteins.

Humans express two active chitinases as well as five chitin-binding proteins that recognize chitin across many tissues (*Bussink et al., 2007*). Chitin levels can be potentially important for mammalian lung and gastrointestinal health. These tissues have distinct pH, with the lung environment normally ~pH 7.0 and the stomach environment normally ~pH 2.0, which raises the question of how chitin-response machinery has evolved to function optimally across such diverse chemical environments. Acidic Mammalian Chitinase (AMCase, also known as Chia, for chitinase, acidic) was originally discovered in the stomach and named for its acidic isoelectric point. AMCase is also constitutively expressed in the lungs at low levels and overexpressed upon chitin exposure (*Van Dyken and Locksley, 2018*; *Zhu et al., 2004*; *Reese et al., 2007*), suggesting this single enzyme has evolved to perform its function under vastly different chemical conditions. Chitin clearance is particularly important for mammalian pulmonary health, where exposure to and accumulation of chitin can be deleterious. In the absence of AMCase, chitin accumulates in the airways, leading to epithelial stress, chronic activation of type 2 immunity, and age-related pulmonary fibrosis (*Van Dyken et al., 2017*; *Van Dyken and Locksley, 2018*).

AMCase is a member of the glycosyl hydrolase family 18 (GH18) (*Davies and Henrissat, 1995*), the members of which hydrolyze sugar linkages through a conserved two-step mechanism where the glycosidic oxygen is protonated by an acidic residue and a nucleophile adds into the anomeric carbon leading to elimination of the hydrolyzed product (*Figure 1A*). This mechanism is corroborated by structures of different GH18 chitinases, most notably *S. marcescens* Chitinase A (PDB ID: 1FFQ) (*Papanikolau et al., 2003*). In inhibitor-bound structures for human AMCase (hAMCase; PDB ID: 3FY1), interactions mimicking the retentive, post-cleavage intermediate state pre-hydrolysis of the oxazolinium intermediate are adopted by the nonhydrolyzable analogs (*Cole et al., 2010*; *Olland et al., 2009*). Unlike the nonhydrolyzable inhibitors, we expect that the oxazolinium intermediate formed from chitin will reopen into the reducing-end GlcNAc monomer unit upon the nucleophilic addition of water.

Biochemical studies of mouse AMCase (mAMCase) measuring relative activity levels demonstrated a global maximum activity at acidic pH, but also a broad second local optimum near neutral pH (*Boot et al., 2001*). This result suggested that mAMCase exhibits two distinct pH optima, which is unlike most enzymes that exhibit a shift or broadening of enzymatic activity across conditions (*Yoong et al., 2006*; *Sajedi et al., 2005*; *Bhunia et al., 2011*). For mAMCase the global maximum near pH 2.0 resembles the chemical environments of the stomach and the local maximum near pH 7.0 is similar to the environment of the lung. These two pH optima in the same enzyme suggest that mAMCase may employ different mechanisms to perform its function in different environments (*Seibold et al., 2009*). In contrast, the human homolog has maximal activity at pH 4.6 with sharply declining activity at more acidic and basic pH (*Seibold et al., 2009*; *Chou et al., 2006*). This optimum corresponds with the pH of lung tissue in pulmonary fibrosis and other disease contexts, suggesting that hAMCase may have been selected for its ability to clear chitin from the lungs and restore healthy lung function.

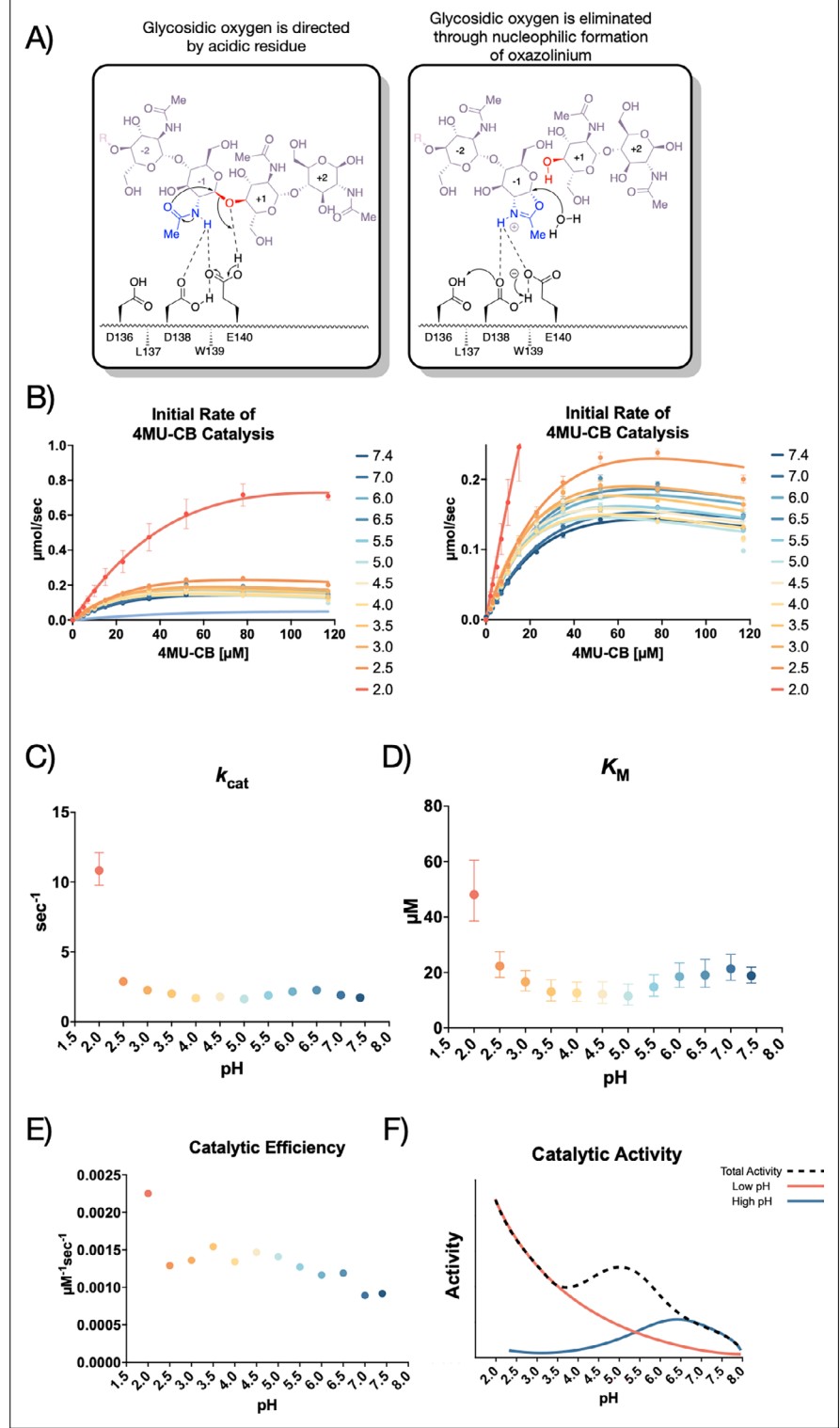

**Figure 1.** Kinetic properties of mAMCase catalytic domain at various pH. (**A**) Chemical depiction of the conserved two-step mechanism where the glycosidic oxygen is protonated by an acidic residue and a nucleophile adds into the anomeric carbon leading to elimination of the hydrolyzed product. (**B**) The rate of 4MU-chitobioside catalysis (1 /s) by mAMCase catalytic domain is plotted as a function of 4MU-chitobioside concentration (μM). Each data point represents n=4 with error bars representing the standard deviation. Michaelis-Menten equation without substrate inhibition was used to estimate the $k_{cat}$ and $K_M$ from the initial rate of reaction at various substrate concentrations. (**C**) The rate of substrate turnover (1 /s) by mAMCase catalytic domain is plotted as a function of

*Figure 1 continued on next page*

*Figure 1 continued*

pH. Error bars represent the 95% confidence interval. (**D**) The Michaelis-Menten constant of mAMCase catalytic domain is plotted as a function of pH. Error bars represent the 95% confidence interval. (**E**) The catalytic efficiency ($k_{cat}/K_M$) of mAMCase catalytic domain is plotted as a function of pH. (**F**) Hypothetical catalytic activity modeled explained by a low pH mechanism (red), and high pH mechanism (blue) and their corresponding total activity (dashed line).

The online version of this article includes the following figure supplement(s) for figure 1:

**Figure supplement 1.** pH of reaction solution before and after quenching with 0.1 M Gly-NaOH pH 10.7.

**Figure supplement 2.** Kinetics of 4MU-chitobioside catalysis by mAMCase catalytic domain at various pH.

The activity of mAMCase has been previously measured through endpoint experiments with limited insight into the rate of catalysis, substrate affinity, and potential substrate inhibition (*Seibold et al., 2009*). While the pH profile of mAMCase has been reported as a percentage of maximum activity at a given pH, it is unclear how the individual kinetic parameters ($K_M$ or $k_{cat}$) vary (*Boot et al., 2001*). These gaps have made it challenging to define the mechanism by which mAMCase shows distinct enzymatic optima at different pHs. One possibility is that mAMCase undergoes structural rearrangements to support this adaptation. Alternatively AMCase may have subtly different mechanisms for protonating the catalytic glutamic acid depending on the environmental pH.

In this work, we explore these hypotheses by employing biophysical, biochemical, and computational approaches to observe and quantify mAMCase function at different pHs. We measured the mAMCase hydrolysis of chitin, which revealed significant activity increase under more acidic conditions compared to neutral or basic conditions. To understand the relationship between catalytic residue protonation state and pH-dependent enzyme activity, we calculated the theoretical pKa of the active site residues and performed molecular dynamics (MD) simulations of mAMCase at various pHs. We also directly observed conformational and chemical features of mAMCase between pH 4.74 and 5.60 by solving X-ray crystal structures of mAMCase in complex with oligomeric GlcNAcn across this range. Together these data support a model in which mAMCase employs two different mechanisms for obtaining a proton in a pH-dependent manner, providing a refined explanation as to how this enzyme recognizes its substrate in disparate environments.

## Results

### New assay confirms broad pH profile for mAMCase

Prior studies have focused on relative mAMCase activity at different pH (*Boot et al., 2001*; *Seibold et al., 2009*; *Kashimura et al., 2015*), limiting the ability to define its enzymological properties precisely and quantitatively across conditions of interest. To expand upon these previous observations of dual optima in mAMCase activity at pH 2.0 and 7.0, we measured mAMCase activity in vitro. We developed an approach that would enable direct measurement of $k_{cat}$ and $K_M$ for mAMCase across a broad pH range by modifying a prior assay that continuously measures mAMCase-dependent breakdown of a fluorogenic chitin analog, 4-methylumbelliferone (4MU) conjugated chitobioside. To overcome the pH-dependent fluorescent properties of 4MU-chitobioside, we reverted the assay into an endpoint assay, which allowed us to measure substrate breakdown across different pH (*Barad et al., 2020*; *Figure 1—figure supplement 1*).

We conducted our endpoint assay across a pH range of 2.0–7.4 to reflect the range of physiological conditions at its in vivo sites of action (*Figure 1B*; Data available at doi: 10.5281/zenodo.8250616). We then derived the Michaelis-Menten parameters at each pH value measured (*Figure 1—figure supplement 2*; Data available at doi: 10.5281/zenodo.8250616). We found that mAMCase has maximum activity at pH 2.0 with a secondary local maximum at pH 6.5, pointing to a bimodal distribution of activity across pH. This is consistent with the relative activity measurements previously performed on mAMCase, but distinct from a single broad pH range, as has been observed for $k_{cat}$ of hAMCase (*Boot et al., 2001*; *Seibold et al., 2009*). The two maxima at pH 2.0 and 6.5 are an approximate match the pH at the primary in vivo sites of mAMCase expression, the stomach and lungs, respectively (*Seibold et al., 2009*). These observations raise the possibility that mAMCase, unlike other AMCase homologs, may have evolved an unusual mechanism to accommodate multiple physiological conditions.

We also found that low pH primarily improves the rate of mAMCase catalysis 6.3-fold ($k$cat; *Figure 1C*), whereas $K$M (*Figure 1D*) worsens 2.5-fold from pH 7.4 to pH 2.0. Similar to chitotriosidase the other active chitinase in mammals and also a GH18 chitinase, we observe an apparent reduction in the rate of mAMCase catalysis across all pH values measured at 4MU-chitobioside concentrations above 80 µM, which suggests that mAMCase may be subject to product inhibition (*Aguilera et al., 2003*). The underlying mechanism for the observed product inhibition may be that mAMCase can transglycosylate the products, as has been previously observed at pH 2.0 and 7.0 (*Wakita et al., 2017*). This potential product inhibition leads to a systematic underprediction of rates by the Michaelis-Menten model at high substrate concentrations. The catalytic efficiency ($k$cat/$K$M) of mAMCase may not capture the effects of product inhibition given that these constants reflect sub-saturating substrate concentrations. Independent of the potential for product inhibition, the trend that mAMCase has highest kcat at very low pH and another local optimum at more neutral pH is clear. We hypothesize that these activity data resemble two overlapping activity distributions, suggesting that the rate at lower pH activity is dependent on the concentration of free protons in solution and that the higher pH optimum results from a distinct mechanism (*Figure 1E*).

## Characterization of mAMCase ligand occupancy and conformational heterogeneity

Our biochemical analyses led us to hypothesize that the pH-dependent activity profile of mAMCase is linked to the mechanism by which catalytic residues are protonated. Previous structural studies on AMCase have focused on interactions between inhibitors like methylallosamidin and the catalytic domain of the protein. We built on these efforts by solving the structure of mAMCase in complex with oligomeric GlcNAcn, the building block of chitin. We used chitin oligomers because they are chemically identical to polymeric chitin found in nature but are soluble and therefore more amenable for co-crystallization than crystalline chitin is. We successfully determined high resolution X-ray crystal structures of the apo mAMCase catalytic domain at pH 5.0 and 8.0 (PDB ID: 8FG5, 8FG7) and holo mAMCase catalytic domain between pH 4.74–5.60 in complex with either $GlcNAc_2$ or GlcNAc3 (PDB ID: 8GCA, 8FRC, 8FR9, 8FRB, 8FRD, 8FRG, 8FRA; *Figure 2—figure supplement 1*; *Table 1*).

Across these different datasets, we observed complex ligand density in the active site of mAMCase. In all of our datasets, we observed continuous ligand density that resembled higher order chitin oligomers (e.g. $GlcNAc_4$, $GlcNAc_5$, or $GlcNAc_6$). This observation was confusing given that these structures were co-crystallized with either $GlcNAc_2$ or $GlcNAc_3$. For example, due to the continuous nature of ligand density observed in our mAMCase-$GlcNAc_3$ co-crystal structure at pH 4.74 (PDB ID: 8GCA, chain A), we initially modeled hexaacetyl-chitohexaose (H-(GlcNAc)6-OH) into the –4 to +2 sugar-binding subsites, using the nomenclature for sugar-binding subsites from *Davies et al., 1997*. This nomenclature defines the sugar-binding subsites as -$n$ to +$n$, with -$n$ corresponding to the non-reducing end and +$n$ the reducing end.

We next continued with a modeling approach that replaced higher order oligomer models with models that only used the chemically defined oligomers present in the crystallization drop. To accomplish this modeling of different binding poses, we placed multiple copies of these oligomers consistent with an interpretation of extensive conformational heterogeneity (*Figure 2—figure supplement 2*). In one sample co-crystallized with $GlcNAc_3$ at pH 4.74 (PDB ID: 8GCA, chains A-B), we identified ligand density that was consistent with $GlcNAc_2$, suggesting that some hydrolysis occurs in the crystal. The resulting model includes compositional heterogeneity as there are both types of oligomer present.

Therefore, across all of our datasets, we modeled a combination of ligand binding events consisting of overlapping $GlcNAc_2$ or $GlcNAc_3$ molecules at each sugar-binding site, i.e. $GlcNAc_2$ ResID 401 Conf. A occupied subsites –3 to –2 while $GlcNAc_2$ ResID 401 Conf. C occupied subsites –2 to –1. By providing each ligand molecule with an alternative conformation ID, this allowed both occupancies and B-factors to be refined (*Figure 2A, B and C*; additional details in Methods). Across these different datasets, we observed ligand density for different combinations of occupancy over the –4 to +2 sugar-binding subsites (*Figure 2A*). While modeling chito-oligomers into strong electron density, we observed strong positive difference density between sugar-binding subsites near the C2 *N*-acetyl and the C6' alcohol moieties. Using the non-crystallographic symmetry (NCS) 'ghost' feature in *Coot*, we were then able to observe that the positive difference density between ligand subsites

**Table 1.** Data collection and refinement statistics.
Statistics for the highest resolution shell are shown in parentheses.

| Dataset | Apo at 100 K | Apo at 277 K | Holo with GlcNAc₃ at pH 4.74 | Holo with GlcNAc₂ at pH 4.91 | Holo with GlcNAc₂ at pH 5.08 | Holo with GlcNAc₂ at pH 5.25 | Holo with GlcNAc₂ at pH 5.25 | Holo with GlcNAc₂ at pH 5.43 | Holo with GlcNAc₂ at pH 5.60 |
|---|---|---|---|---|---|---|---|---|---|
| PDB ID | 8FG5 | 8FG7 | 8GCA | 8FRC | 8FR9 | 8FRB | 8FRD | 8FRG | 8FRA |
| Diffraction Data DOI | 10.18430/ M38FG5 | 10.18430/ M38FG7 | 10.18430/ M38GCA | 10.18430/ M38FRC | 10.18430/ M38FR9 | 10.18430/ M38FRB | 10.18430/ M38FRD | 10.18430/M38FRG | 10.18430/M38FRA |
| pH | 5.00 | 8.00 | 4.74 | 4.91 | 5.08 | 5.25 | 5.25 | 5.43 | 5.60 |
| Ligand | N/A | N/A | GlcNAc₃ | GlcNAc₂ | GlcNAc₂ | GlcNAc₂ | GlcNAc₂ | GlcNAc₂ | GlcNAc₂ |
| [Ligand] mM | N/A | N/A | 12.67 | 29.00 | 19.33 | 19.33 | 29.00 | 29.00 | 19.33 |
| Wavelength | 1.117 | 1.116 | 1.116 | 1.116 | 1.116 | 1.116 | 1.116 | 1.116 | 1.116 |
| Resolution range | 46.8–1.3 (1.346–1.3) | 50.88–1.64 (1.699–1.64) | 61.83–1.7 (1.761–1.7) | 69.52–1.92 (1.989–1.92) | 69.59–1.5 (1.554–1.5) | 57.29–1.7 (1.761–1.7) | 58.67–1.68 (1.74–1.68) | 69.59–1.741 (1.803–1.741) | 86.27–1.95 (2.02–1.95) |
| Space group | P 1 2 1 1 | P 21 21 21 | P 21 21 2 | P 2 21 21 | P 2 21 21 | P 21 21 21 | P 2 21 21 | P 21 21 2 | P 21 21 21 |
| Unit cell (length) | 60.04 42.25 67.41 | 63.6466 71.8436 84.6724 | 76.0664 91.7195 106.132 | 70.9333 92.6896 105.123 | 71.1131 92.6412 105.423 | 91.9263 106.963 146.492 | 70.755 92.451 104.99 | 92.8934 105.041 70.8116 | 92.0659 106.705 146.57 |
| Unit cell (angles) | 90 95.18 90 | 90 90 90 | 90 90 90 | 90 90 90 | 90 90 90 | 90 90 90 | 90 90 90 | 90 90 90 | 90 90 90 |
| Total reflections | 2099252 (194837) | 620486 (61796) | 516529 (48842) | 339863 (33874) | 702566 (63651) | 1010525 (98078) | 499250 (48902) | 420425 (37138) | 691049 (67775) |
| Unique reflections | 83050 (8251) | 47999 (4678) | 82111 (8079) | 53587 (5242) | 109106 (10560) | 158679 (15679) | 78153 (7593) | 71329 (6974) | 105512 (10401) |
| Multiplicity | 25.3 (23.6) | 12.9 (13.2) | 6.3 (6.0) | 6.3 (6.5) | 6.4 (6.0) | 6.4 (6.3) | 6.4 (6.4) | 5.9 (5.3) | 6.5 (6.6) |
| Completeness (%) | 99.99 (99.98) | 99.37 (98.65) | 99.72 (99.42) | 99.88 (99.79) | 97.48 (95.47) | 99.87 (99.88) | 98.71 (97.03) | 99.56 (99.03) | 99.74 (99.62) |
| Mean I/sigma(I) | 13.31 (1.88) | 7.00 (1.19) | 8.83 (3.12) | 7.72 (3.21) | 16.77 (5.46) | 9.09 (3.10) | 9.68 (3.09) | 6.18 (2.56) | 5.65 (1.26) |
| Wilson B-factor | 15.81 | 16.38 | 12.17 | 13.44 | 9.16 | 12.47 | 11.55 | 15.76 | 12.64 |
| R-merge | 0.1342 (2.107) | 0.2489 (2.119) | 0.1811 (1.138) | 0.1531 (0.5265) | 0.06539 (0.2976) | 0.1111 (0.5593) | 0.1155 (0.569) | 0.1321 (0.4674) | 0.1619 (0.6276) |
| R-meas | 0.137 (2.153) | 0.2591 (2.203) | 0.1972 (1.242) | 0.1669 (0.5728) | 0.07122 (0.3259) | 0.121 (0.61) | 0.126 (0.6197) | 0.1448 (0.5188) | 0.176 (0.6822) |

*Table 1 continued on next page*

*Table 1 continued*

| Dataset | Apo at 100 K | Apo at 277 K | Holo with GlcNAc$_3$ at pH 4.74 | Holo with GlcNAc$_2$ at pH 4.91 | Holo with GlcNAc$_2$ at pH 5.08 | Holo with GlcNAc$_2$ at pH 5.25 | Holo with GlcNAc$_2$ at pH 5.25 | Holo with GlcNAc$_2$ at pH 5.43 | Holo with GlcNAc$_2$ at pH 5.60 |
|---|---|---|---|---|---|---|---|---|---|
| R-pim | 0.02718 (0.4382) | 0.07097 (0.5968) | 0.07709 (0.4917) | 0.06573 (0.2233) | 0.02784 (0.1311) | 0.04745 (0.2411) | 0.04965 (0.2425) | 0.05834 (0.2207) | 0.06836 (0.2647) |
| CC1/2 | 0.999 (0.858) | 0.996 (0.502) | 0.997 (0.805) | 0.994 (0.884) | 0.999 (0.943) | 0.997 (0.888) | 0.993 (0.68) | 0.994 (0.845) | 0.997 (0.845) |
| CC* | 1 (0.961) | 0.999 (0.818) | 0.999 (0.944) | 0.998 (0.969) | 1 (0.985) | 0.999 (0.97) | 0.998 (0.9) | 0.998 (0.957) | 0.999 (0.957) |
| Reflections used in refinement | 83046 (8251) | 47968 (4677) | 82030 (8059) | 53543 (5242) | 109065 (10557) | 158531 (15678) | 78103 (7592) | 71295 (6967) | 105380 (10401) |
| Reflections used for R-free | 4099 (422) | 2328 (234) | 4142 (427) | 2738 (273) | 5449 (559) | 7978 (802) | 3878 (334) | 3561 (348) | 5174 (542) |
| R-work | 0.1317 (0.2361) | 0.1469 (0.2707) | 0.1598 (0.2428) | 0.1472 (0.1616) | 0.1376 (0.1615) | 0.1423 (0.1850) | 0.1396 (0.1724) | 0.1657 (0.2194) | 0.1695 (0.2074) |
| R-free | 0.1519 (0.2613) | 0.1717 (0.3244) | 0.1978 (0.2952) | 0.1898 (0.2065) | 0.1644 (0.1932) | 0.1778 (0.2315) | 0.1689 (0.2113) | 0.2083 (0.2737) | 0.2056 (0.2463) |
| CC(work) | 0.970 (0.583) | 0.978 (0.789) | 0.969 (0.819) | 0.953 (0.846) | 0.971 (0.922) | 0.970 (0.878) | 0.963 (0.903) | 0.959 (0.749) | 0.961 (0.869) |
| CC(free) | 0.969 (0.558) | 0.975 (0.729) | 0.953 (0.775) | 0.951 (0.793) | 0.966 (0.910) | 0.958 (0.791) | 0.954 (0.882) | 0.951 (0.757) | 0.970 (0.846) |
| Number of non-hydrogen atoms | 3583 | 3427 | 7330 | 6953 | 7507 | 13986 | 6951 | 7343 | 14428 |
| macromolecules | 3107 | 3097 | 6094 | 6016 | 6186 | 11938 | 6019 | 6286 | 11900 |
| ligands | 1 | 1 | 394 | 342 | 516 | 746 | 344 | 401 | 571 |
| solvent | 475 | 329 | 1034 | 763 | 1057 | 1666 | 756 | 852 | 2237 |
| Protein residues | 376 | 376 | 752 | 738 | 750 | 1478 | 738 | 738 | 1478 |
| Nucleic acid bases | | | | | | | | | |
| RMS(bonds) | 0.006 | 0.008 | 0.008 | 0.007 | 0.01 | 0.006 | 0.007 | 0.008 | 0.003 |
| RMS(angles) | 0.88 | 0.96 | 1.05 | 0.91 | 1.1 | 0.92 | 0.91 | 1.12 | 0.66 |
| Ramachandran favored (%) | 98.4 | 98.66 | 98.8 | 98.23 | 98.26 | 98.84 | 98.64 | 98.35 | 98.1 |
| Ramachandran allowed (%) | 1.6 | 1.34 | 1.2 | 1.77 | 1.74 | 1.16 | 1.36 | 1.65 | 1.9 |

*Table 1 continued on next page*

*Table 1 continued*

| Dataset | Apo at 100 K | Apo at 277 K | Holo with GlcNAc$_3$ at pH 4.74 | Holo with GlcNAc$_2$ at pH 4.91 | Holo with GlcNAc$_2$ at pH 5.08 | Holo with GlcNAc$_2$ at pH 5.25 | Holo with GlcNAc$_2$ at pH 5.25 | Holo with GlcNAc$_2$ at pH 5.43 | Holo with GlcNAc$_2$ at pH 5.60 |
|---|---|---|---|---|---|---|---|---|---|
| Ramachandran outliers (%) | 0 | 0 | 0 | 0 | 0 | 0 | 0 | 0 | 0 |
| Rotamer outliers (%) | 1.22 | 0.92 | 0.62 | 0.79 | 0.92 | 0.87 | 0.63 | 0.6 | 0.88 |
| Clashscore | 1.66 | 0.83 | 1.25 | 1.85 | 1.3 | 1.31 | 1.6 | 1.44 | 1.66 |
| Average B-factor | 21.71 | 19.1 | 16.09 | 14.55 | 12.73 | 15.72 | 14.2 | 17.9 | 15.9 |
| macromolecules | 19.83 | 17.9 | 13.9 | 13.24 | 10.3 | 13.76 | 12.5 | 16.36 | 13.88 |
| ligands | 98.88 | 46.35 | 23.57 | 18.87 | 15.73 | 17.53 | 15.9 | 23.5 | 19.18 |
| solvent | 33.82 | 30.3 | 27.53 | 23.9 | 26.25 | 29.3 | 27.32 | 27.98 | 26.24 |
| Number of TLS groups | | | | | | | | | |

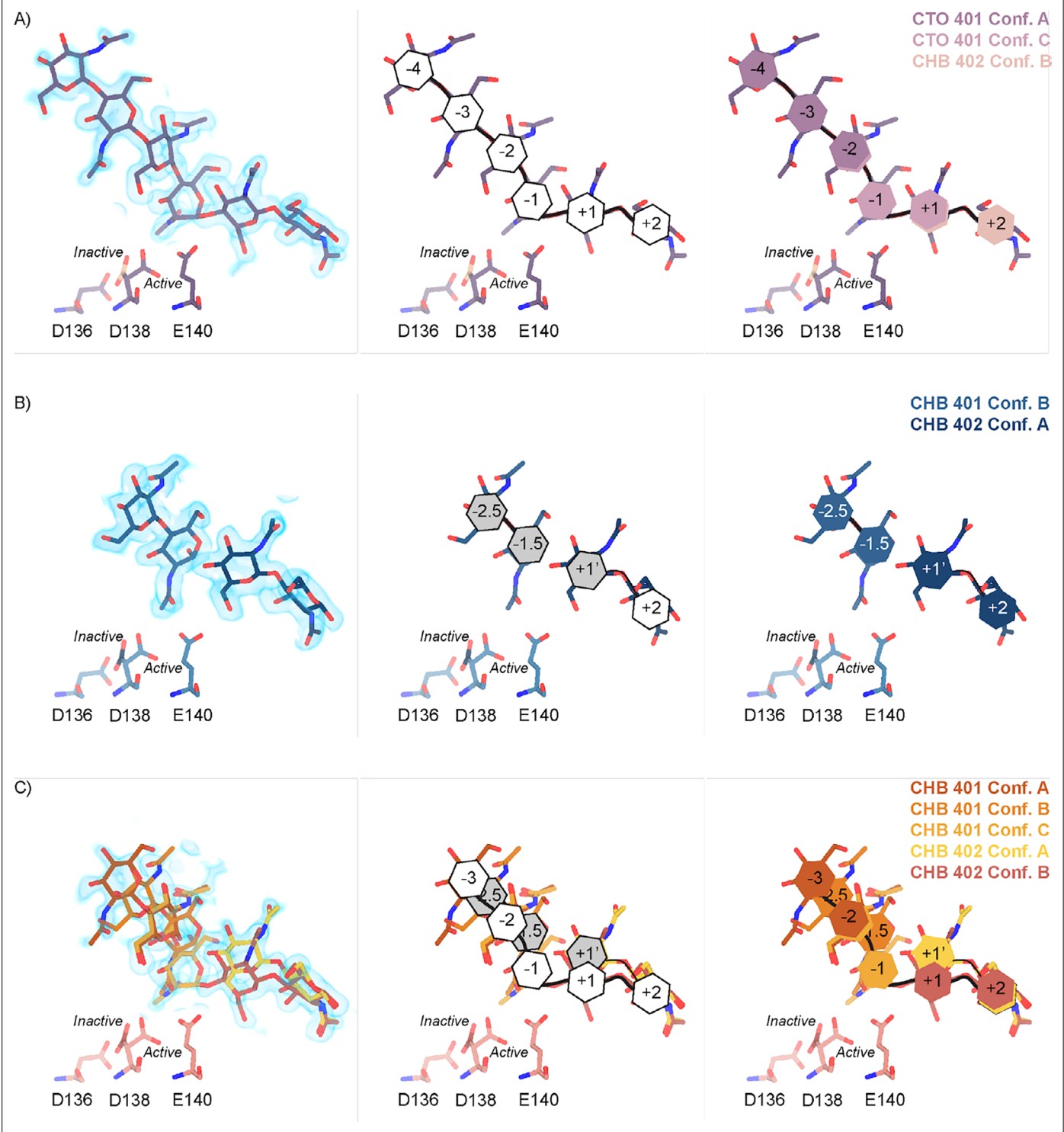

**Figure 2.** Schematic representation of sugar-binding subsites in mAMCase. (**A**) PDB ID: 8GCA, chain A. Stick representation of all GlcNAc₂ sugar-binding events observed in *n* sugar-binding subsites with 2mFo-DFc map shown as a 1.2 Å contour (blue), the subsite nomenclature, and a schematic of alternative conformation ligand modeling. (**B**) PDB ID: 8FRA, chain D. Stick representation of all GlcNAcₙ binding events observed in n+0.5 sugar-binding subsites with 2mFo-DFc map shown as a 1.2 Å contour (blue), the subsite nomenclature, and a schematic of alternative conformation ligand modeling. (**C**) PDB ID: 8FR9, chain B. Stick representation of all GlcNAcₙ binding events observed in *n* and n+0.5 sugar-binding subsites with 2mFo-DFc map shown as a 1.2 Å contour (blue), the subsite nomenclature, and a schematic of alternative conformation ligand modeling.

The online version of this article includes the following figure supplement(s) for figure 2:

*Figure 2 continued on next page*

*Figure 2 continued*

**Figure supplement 1.** 96-well plate layout of crystallization conditions.

**Figure supplement 2.** pKa of apo and holo mAMCase in the D2 *inactive* and *active* conformation.

**Figure supplement 3.** Overview of key residues for mAMCase activity.

**Figure supplement 4.** Protein-ligand interactions between mAMCase and chitin.

**Figure supplement 5.** Ringer analysis of catalytic triad confirms alternative Asp138 conformations.

in one chain could be explained by the dominant ligand pose observed in another associated crystallographic chain, suggesting the presence of a low-occupancy binding events. This observation led to the discovery that GlcNAcn occupies intermediate subsites, which we label n+0.5, continuing to follow the nomenclature established by Davies et al., in addition to canonical sugar-binding subsites (*Figure 2B*; *Davies et al., 1997*).

In addition to identifying novel n+0.5 sugar-binding subsites, we also observed strong positive difference density above the +1 subsite, which we label +1'. During ligand refinement, we observed density for both the α- and β–1,4-linked $GlcNAc_2$ anomers in the active site. This unexpected configurational heterogeneity, which is observable because of the high resolution of our datasets (1.30–1.95 Å), likely formed as a result of equilibration between the two anomers through an oxocarbenium close-ion-pair intermediate. The ability for the active site to accommodate and form interactions with these ligands is important given its role in degrading crystalline chitin, a complex and often recalcitrant substrate that likely requires multiple binding events by AMCase before degradation can occur. We did not identify consistent trends between the contents of the crystallization drop (pH, substrate identity, and substrate concentration), the crystal properties (space group, unit cell dimensions, resolution), and the resulting density in the active site; however, as outlined below, the protein conformations and substrate states are highly correlated. Collectively, modeling a combination of ligand binding modes, linkages, and anomers allowed us to interpret the resulting coordinates in a more complete model of how mAMCase coordinates and stabilizes polymeric chitin for catalysis (*Figure 2*; *Figure 2—figure supplement 2*; *Figure 2—figure supplement 3*; *Supplementary file 1*).

## Structural characterization of mAMCase catalytic triad $D_1xD_2xE$

We interpreted the protein-ligand interactions along the canonical binding sites (*Figure 2—figure supplement 2*). As with other chitinases, we observe a network of tryptophans consisting of Trp31, Trp360, Trp99, and Trp218 stabilizing the positioning of the ligand into the binding site through a series of H-π interactions with the −3,−1,+1, and +2 sugars, respectively (*Watanabe et al., 2003*; *Horn et al., 2006*; *Zakariassen et al., 2009*). These interactions are primarily with the axial hydrogens of the respective sugars but also include the N-H of the −3 and +1 sugar and the 6' O-H of the +2 sugar (*Figure 2—figure supplement 3*). Further, we observe Asp213 accepting a hydrogen bond with the 6' OH of the −1 sugar and Tyr141 acting as a hydrogen bond donor to the 6' OH of the +1 sugar. These two hydrogen bonds likely orient the ligand in the catalytically competent pose where the glycosidic oxygen bridging the −1 and +1 sugars is 2.8 Å away from the acidic Glu140 -OH (*Figure 2—figure supplement 4*). With this proximity, Glu140 can act as a hydrogen bond donor to the strained (122o bond angle) bridging oxygen forming a hydrogen bond to promote the formation of an oxazolinium intermediate and subsequent cleavage of the glycosidic bond. We observed two interactions with the sugar in the −4 position supporting the ligand orientation far from the enzymatic active site. Residues involved in ligand binding and catalysis adopt similar side chain conformations in the absence of ligand (PDB ID: 8FG5, 8FG7), suggesting that the active site is organized prior to ligand binding and not subject to ligand-stabilized conformational changes.

We hypothesize that the +1' subsite is primarily occupied by the product $GlcNAc_2$ prior to its displacement from the active site by subsequent sliding of polymeric chitin (*Figure 2B*; *Jiménez-Ortega et al., 2021*). At this position, Trp99 and Trp218 engage in CH-π interactions with the +1 and+2 sugars, respectively while Asp213 forms a new H-bond with the carbonyl oxygen and Tyr141 retains an H-bond with the hydroxyl moiety on the +1 sugar. We are able to observe this post-catalysis binding mode due to the stabilizing interactions between $GlcNAc_2$ and Asp213, Trp99, Trp218, and Tyr141 (*Figure 2—figure supplement 3*). Together, these observations highlight the dynamic chitin binding modes within the mAMCase active site. Collectively, the observed non-canonical binding modes of

these sugars is consistent with previous observations that once bound to polymeric chitin, GH18 chitinases engage in chain sliding from the reducing end of the substrate following catalysis (*Nakamura et al., 2018*).

In contrast to the largely static interactions outlined above, we observed conformational heterogeneity in the catalytically critical Asp138 residue, suggesting flipping between two equally stable states facing each of the other two residues in the catalytic triad (Asp136 or Glu140; *van Aalten et al., 2001*). Using Ringer, we confirmed that there are two Asp138 conformations and only a single conformation for Asp136 and Glu140 (*Figure 2—figure supplement 4*; Data available at doi: 10.5281/zenodo.7758815; *Lang et al., 2010*). Across 20 chains from the datasets derived from different pH and co-crystallization conditions (*Supplementary file 1*), we quantified whether Asp138 is preferentially oriented towards Asp136 (*inactive* conformation) or preferentially oriented towards Glu140 (*active* conformation).

Prior work has suggested that Asp138 orients itself towards Glu140 to promote stabilization of the substrate's twisted boat conformation in the −1 subsite. Therefore, we explored if Asp138 conformation is correlated with ligand pose (*Olland et al., 2009*; *van Aalten et al., 2001*; *Fusetti et al., 2002*; *Songsiriritthigul et al., 2008*). As previously mentioned, we assign alternative conformation IDs to each ligand molecule based on its subsite positioning. We calculate subsite occupancy by taking the sum of all alternative ligand conformations at a given subsite, i.e. the occupancy of subsite −2 is equal to the occupancies of GlcNAc$_2$ ResID 401 Conf. A and GlcNAc2 ResID 401 Conf. C (*Figure 3A*; see Methods for additional details; Data available at doi: 10.5281/zenodo.7905828). We observe a strong positive correlation between Asp138 conformation and ligand pose only in the −2 to +1 subsites (*Figure 3B*; *Supplementary file 1*). When the −1 subsite is at least 50% occupied, Asp138 prefers the *active* conformation (up towards Glu140). In this orientation, Asp138(HD2) forms a H-bond with Glu140(OE1) (2.6 Å) while Asp138(OD1) forms an H-bond with the amide nitrogen of GlcNAc in the −1 subsite (2.6 Å). Glu140(OE2) is 2.8 Å away from the glycosidic oxygen bridging the −1 and +1 sugars. We suspect that the inverse correlation between Asp138 *active* conformation and the −2.5 and −1.5 sugar-binding subsites represents ligand translocation toward the catalytic residues, prior to enzyme engagement with the ligand. When chitin occupies a canonical sugar-binding subsite, AMCase forms stabilizing H-bonds with the ligand prior to catalysis. These observations are consistent with the proposed catalytic mechanism where upon protonation, the equilibrium between Asp138 conformations shifts to favor the *active* conformation (toward Glu140) where Asp138 stabilizes Glu140 in proximity to the glycosidic oxygen prior to catalysis.

## Theoretical pKa calculations of mAMCase catalytic triad D$_1$xD$_2$xE

Based on the dual pH optimum observed in our kinetics assay and the conformational heterogeneity of Asp138, we calculated the theoretical pKa for catalytic D1xD2xE motif on mAMCase using PROPKA 3.0. PROPKA does not account for alternative conformations in its calculations, so we split our protein models to contain single conformations of the catalytic residues Asp136, Asp138, and Glu140. While PROPKA does account for ligands in its calculations, running the calculations with different alternative conformations of GlcNAc$_2$ or GlcNAc$_3$ had little effect on the calculated pKas for the active site residues (*Figure 2—figure supplement 2*; Data available at doi: 10.5281/zenodo.7905863). Despite the observed ligand heterogeneity, we observe a relatively narrow range of pKa values for the catalytic triad. This suggests that the pKa of the catalytic residues is primarily influenced by the position of nearby residues and that the placement of solvent or ligand molecules has little effect. When Asp138 is oriented towards Asp136 (*the inactive* conformation), the pKa of the catalytic residues are 2.0, 13.0, 7.7 for Asp136, Asp138, and Glu140 respectively. Similarly, when Asp138 is oriented towards Glu140 (*the active* conformation), the pKa of the catalytic residues are 3.4, 12.4, 6.4 for Asp136, Asp138, and Glu140, respectively. Taking this information together, it is clear that the pKa of Asp136 and Glu140 are both affected by the orientation of Asp138 (*Figure 4A*; *Supplementary file 2*; Data available at doi: 10.5281/zenodo.7905863). The pKa of Asp136 suggests that at pH >3.4, Asp136 is deprotonated, and its conjugate base is more stable. We observe a similar pKa distribution for the catalytic triad in human AMCase and other GH18 chitinases with publicly available structures and optimum pH activity profiles (*Figure 4A–C*).

Given the pH range of our crystallization conditions, we expect that Asp136 is deprotonated while Asp138 and Glu140 are protonated. We hypothesize that this anionic aspartate is capable of forming

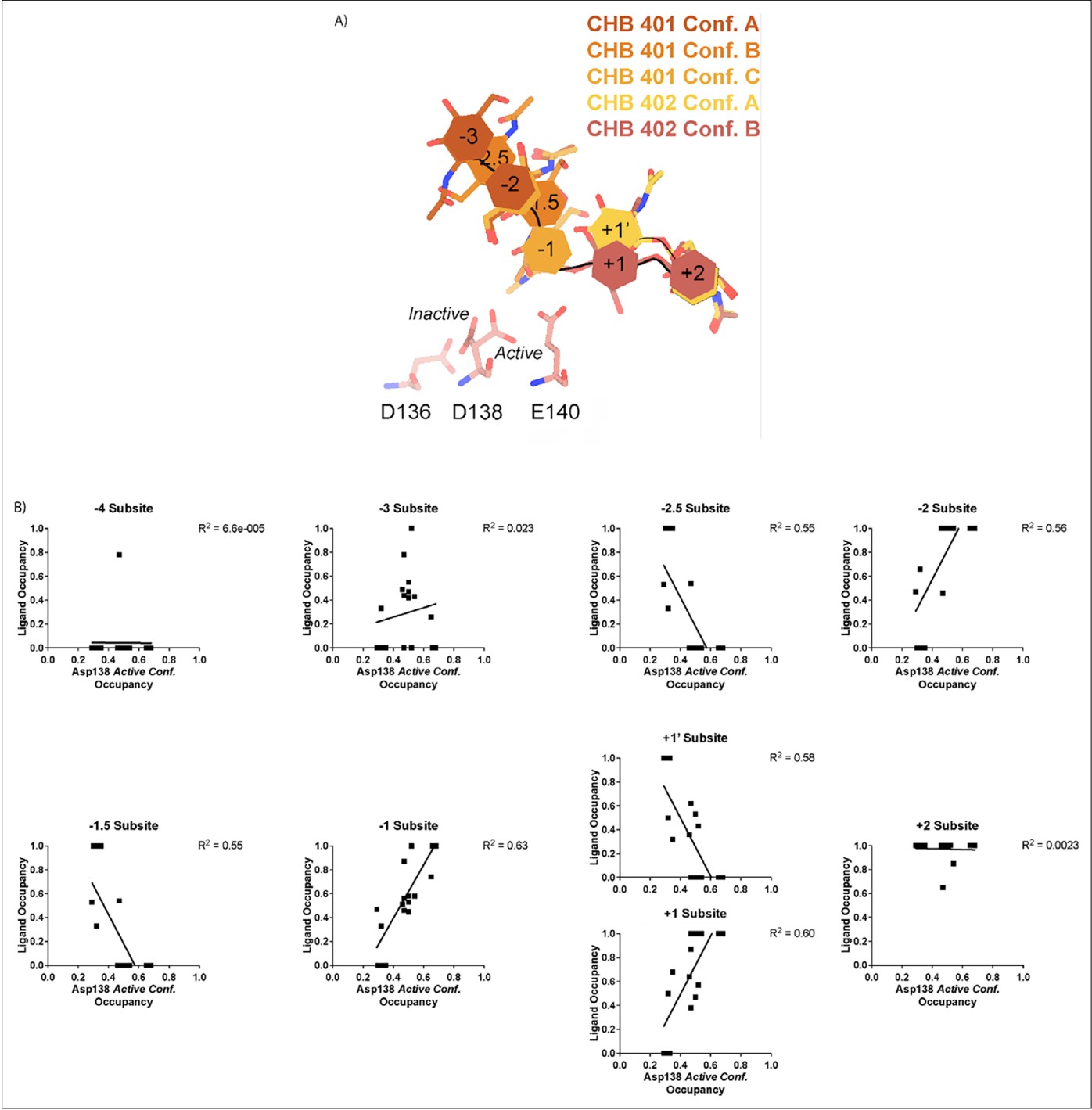

**Figure 3.** Asp138 orientation correlates with ligand subsite occupancy. (**A**) PDB ID: 8FR9, chain B. Schematic of the alternative conformation ligand modeling. (**B**) Linear correlation between sugar-binding subsite occupancy and Asp138 *active* conformation occupancy.

a strong ionic hydrogen bond interaction with Asp138 orienting it in the *inactive* conformation. When Asp136 is protonated to its aspartic acid state at pH <3.2, we expect that it is only capable of forming the relatively weaker neutral hydrogen bond with Asp138 lowering the favorability of the *inactive* conformation.

Additionally, when interpreting the pKa of Glu140, we hypothesize that under acidic conditions (pH 2.0–6.5), Glu140 is capable of obtaining its catalytic proton from solution. The accessibility of Asp138's proton to Glu140 progressively decreases as pH increases from pH 2.0–6.5. In contrast,

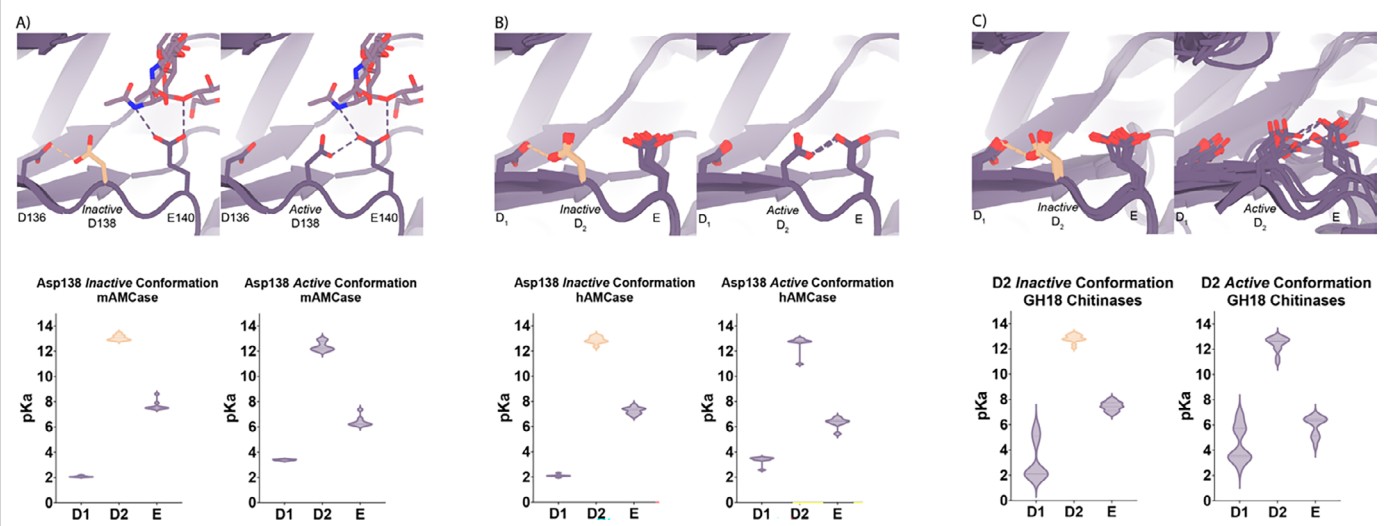

**Figure 4.** pKa of GH18 chitinases in the D2 inactive and active conformation. (**A**) PDB ID: 8GCA, chain A. Distribution of pKa across Asp136, Asp138, Glu140 of mAMCase structures in either Asp138 *inactive* or Asp138 *active* conformation. (**B**) PDB ID: 3FXY, 3RM4, 3RM8, 3RME (*inactive conformation*); 2YBU, 3FY1 (*active conformation*). Distribution of pKa across Asp136, Asp138, Glu140 of hAMCase structures in either Asp138 *inactive* or Asp138 *active* conformation. (**C**) PDB ID: 3ALF, 3AQU, 3FXY, 3RM4, 3RM8, 3RME (*inactive* conformation); 2UY2, 2UY3, 2YBU, 4HME, 4MNJ, 4R5E, 4TXE (*active* conformation). Distribution of pKa across the catalytic triad $D_1xD_2xE$ of GH18 chitinases in either $D_2$ *inactive* or *active* conformation.

under neutral and basic conditions (pH 6.0–7.4), Asp138 can shuttle a proton from Asp136 by rotating about its Cα-Cβ bond to supply Glu140 with the proton. Glu140 subsequently uses the proton that it obtained from Asp138 to protonate the glycosidic bond in chitin, promoting hydrolysis as previously described in several chitinases (*van Aalten et al., 2001*; *Synstad et al., 2004*; *Bussink et al., 2008*). While this mechanism could explain how mAMCase has a local optimum at pH 2.0, it is insufficient to explain why we do not observe a similar optimum in hAMCase. The narrow range of pKa values across GH18 chitinases suggest that differences in optimal activity by pH may be influenced by other factors, such as protein stability, conformational dynamics, or coordination of distal GlcNAc residues by ionizable residues (*Mishra et al., 2021*).

## Molecular dynamics

Based on our enzymology results suggesting the possibility of differential activity between acidic pH (pH 2.0) and near neutral pH (pH 6.5) and theoretical pKa calculations of the active site residues, we performed short atomistic molecular dynamics simulations to interrogate the movement of catalytic residues. While all the crystal structures we obtained were collected in a narrow acidic pH range between 4.74–5.60, we ran simulations at pH 2.0 and pH 6.5, ensuring that the protonation states of side chains populated by 3DProtonate were supported by our PROPKA calculations (Data available at doi: 10.5281/zenodo.7758821; *Labute, 2009*; *Olsson et al., 2011*). These simulations allowed us to investigate our hypothesis that at neutral pH mAMCase enzymatic activity is dependent on the protonation state of Asp136. We performed simulations using protein models that contain Asp138 in either the *inactive* (down towards Asp136; '*inactive* simulation') or *active* conformation (up towards Glu140; '*active* simulation') to avoid bias from the starting conformation.

In all our simulations, we observe that Glu140 orients its acidic proton towards the glycosidic bond between the –1 and +1 sugars. The distance between the acidic proton of Glu140 and the glycosidic oxygen fluctuates between 1.5 and 2.3 Å for the duration of the simulation, with a median distance of 1.8 Å. The positioning of this proton is necessary to allow for the oxocarbenium cleavage of the glycosidic bond and recapitulates the positioning of Glu140 in our experimental structures. In simulations initiated from the *inactive* conformation at pH 2.0, we observe that Asp 138 is readily able to rotate about its Cα-Cβ bond to adopt the *active* conformation forming the same hydrogen bond between Asp138 and Glu140. In contrast, from simulations at pH 6.5 started from the Asp138 *inactive* conformation, we observe that Asp138 remains hydrogen bonded to Asp136 throughout the duration of the simulation (*inactive* conformation; *Figure 5A–C*; Data available at doi: 10.5281/zenodo.7758821). This

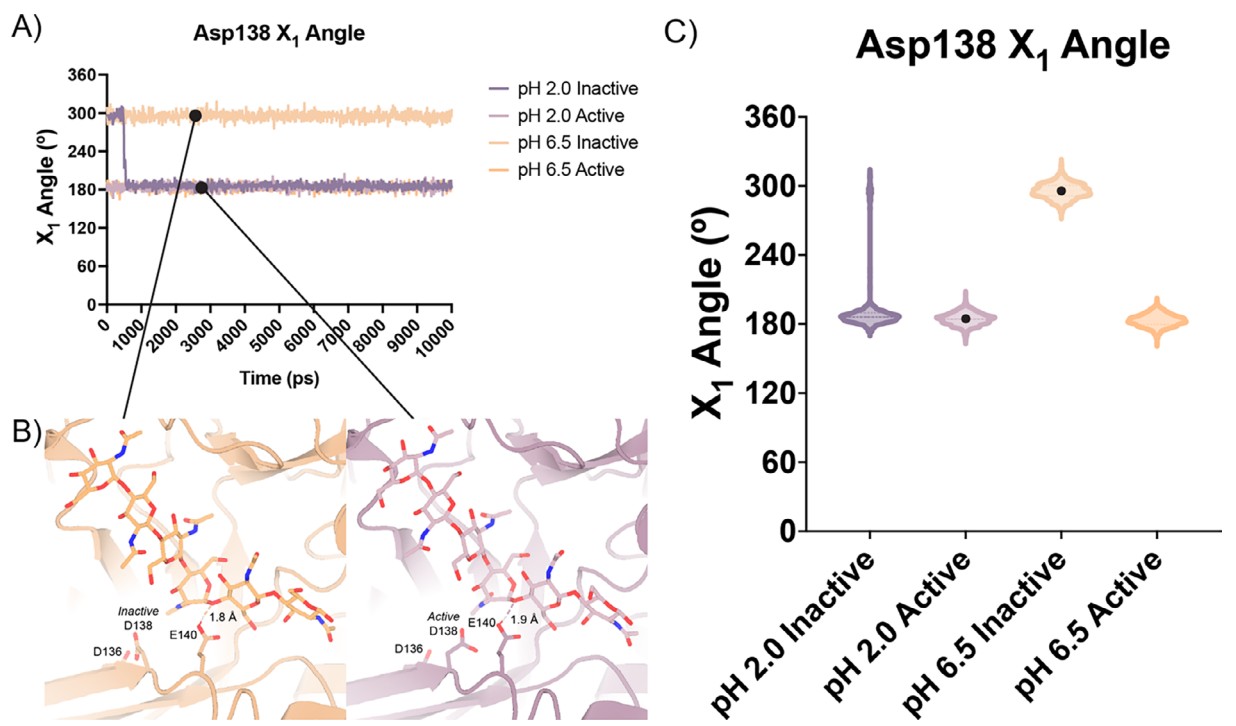

**Figure 5.** Distribution of distances observed every 10 ps of each simulation and their respective time courses. (**A**) Asp138 $\chi_1$ angles over a 10 ns simulation. (**B**) Representative minimum distance snapshots of structure during pH 6.5 *inactive* simulation (left), and pH 2.0 *active* simulation (right). (**C**) Distribution of Asp138 $\chi_1$ angles over a 10 ns simulation.

series of simulations allowed us to better visualize which catalytic side chains are dynamic and which catalytic side chains positioning are well maintained to help build our catalytic mechanism.

## Discussion

mAMCase is an unusual enzyme that can bind and degrade polymeric chitin in very different pH environments. We hypothesized that mAMCase employs different mechanisms to protonate its catalytic glutamate under acidic and neutral pH. Through our analysis, we hypothesize that the observed ligand and catalytic residue densities and occupancies in our crystal structures are consistent with the previously proposed GH18 catalytic mechanism (*Meekrathok et al., 2017*). By modeling GlcNAc$_2$ as sequentially overlapping ligands in alternative conformations (*Figure 2*), we are able to visualize each step in the proposed catalytic cycle of mAMCase (*Figure 6*, *Animation 1*, *Animation 2*). This mechanism, which has been observed in other glycoside hydrolases, occurs when the glycosidic oxygen is protonated by an acidic residue and a nucleophile adds into the anomeric carbon leading to elimination of the hydrolyzed product.

Based on our crystal data and simulations, we envision that at neutral pH, Asp136 is deprotonated (pKa = 2.1) forming an ionic hydrogen bond with Asp138 (pKa = 13.1). In contrast, at low pH Asp136 is protonated, yet continues to form a weaker hydrogen bond with Asp138 (*Figure 6* - Step 1). Glu140 (pKa = 7.7) is protonated across the enzyme's active pH range. Upon ligand binding (*Figure 6* - Step 2), Glu140 stabilizes the sugar at the –1 subsite. The ligand then translocates forward by one GlcNAc$_2$ to occupy the +1 and +2 subsites (*Figure 6* - Step 3). At neutral pH, Asp136 is predominantly deprotonated. When protonation of Asp136 occurs, this destabilizes the Asp136-Asp138 hydrogen bond and allows Asp138 to rotate about its Cα-Cβ bond into the *active* conformation (towards Glu140). However, since Asp136 is always protonated at low pH, the Asp136-Asp138 hydrogen bond is less energetically favorable, therefore Asp138 can adopt the *active* conformation more readily (*Figure 6*- Step 4).

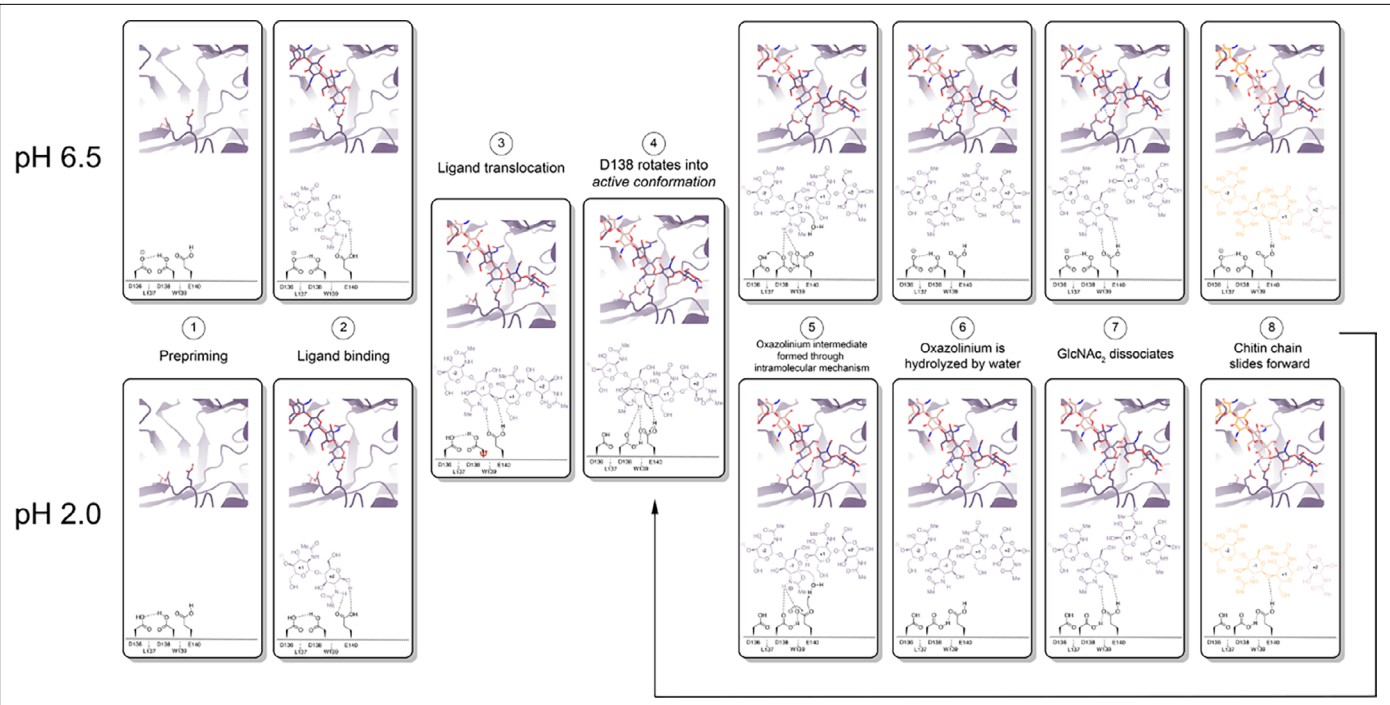

**Figure 6.** Proposed model for ligand translocation towards the active site and ligand release post-catalysis. (**A**) PDB ID: 8GCA, chain A with no ligand (step 1); with GlcNAc$_4$ generated by *phenix.elbow* using PubChem ID: 10985690 (step 2); with GlcNAc$_6$ generated by *phenix.elbow* using PubChem ID: 6918014 (step 3–4, 8); with oxazolinium intermediate generated by *phenix.elbow* using PubChem ID: 25260046 (steps 5.1–5.2); with GlcNAc$_2$ and GlcNAc$_4$ generated by *phenix.elbow* using PubChem ID: 439544 and 10985690, respectively (steps 6–7). Chemical representation of GH18 catalytic cycle with corresponding molecular models of each step. Catalytic residues Asp136, Asp138, Glu140, and ligands are shown as sticks. Protons are shown as gray spheres.

Once Asp138 is in the *active* conformation, Asp138 and Glu140 form stabilizing interactions with the *N*-acetyl group of the ligand, priming it to become the nucleophile required for catalysis (*Figure 6* - Step 4). Glu140 provides its ionizable proton to the ligand's glycosidic oxygen, increasing the electrophilicity of the anomeric carbon (*Figure 6* - Step 5; *Iino et al., 2019*). The carbonyl oxygen of the −1 sugar *N*-acetyl group then nucleophilically adds into the anomeric carbon from the β face to cleave the glycosidic bond, forming the oxazolinium intermediate. At neutral pH, the resultant deprotonated Glu140 is then re-protonated through proton shuttling in which Asp136 donates its proton to Asp138 and Asp138 donates its ionizable proton to Glu140. At acidic pH, we propose that Glu140 can be directly re-protonated by a proton in solution (*Figure 6* - Step 5). At a neutral pH, this leads to Asp138 returning to an *inactive* conformation. However, at low pH Asp136 and Glu140 are both protonated due to the high concentration of protons in solution, allowing

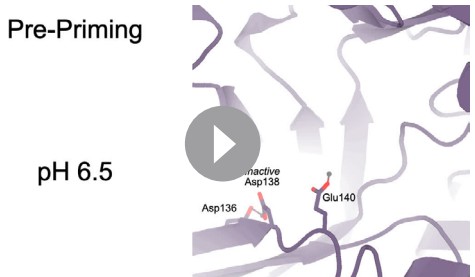

**Animation 1.** Animated movie of the mAMcase catalytic cycle at pH 6.5. Catalytic residues Asp136, Asp138, Glu140, and ligands are shown as sticks. Protons are shown as gray spheres.

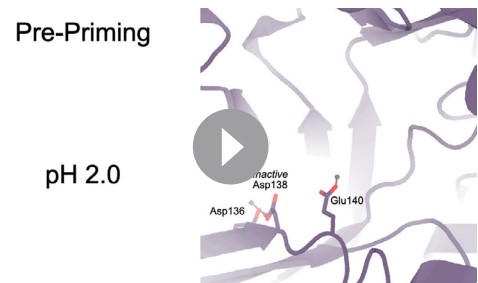

**Animation 2.** Animated movie of the mAMcase catalytic cycle at pH 2.0. Catalytic residues Asp136, Asp138, Glu140, and ligands are shown as sticks. Protons are shown as gray spheres.

Asp138 to remain in the *active* conformation and form stabilizing interactions with the *N*-acetyl group on the ligand. The oxazolinium intermediate is then hydrolyzed by a water molecule, generating a GlcNAc$_2$ catalysis product in the +1 and+2 sugar subsites (*Figure 6* - Step 6). The GlcNAc$_2$ product dissociates from the +1 to+2 sugar subsites, then the ligand undergoes 'decrystallization' and 'chain sliding' before re-entering the catalytic cycle, assuming AMCase is bound to a longer polymer such as its natural substrate (*Nakamura et al., 2018*). At neutral pH this catalytic mechanism is reset with Asp138 in its *inactive* conformation, however at low pH the catalytic mechanism is reset with Asp138 already in the *active* conformation. This could lead to faster rates of catalysis at lower pH compared to the neutral pH mechanism, providing a possible explanation for the observed changes in rate at varying pH.

While our model helps us propose a plausible explanation of why mAMCase is highly active at pH 2, it does not explain why hAMCase has a single activity optimum around pH 5.

Prior work by Kashimura et al. has demonstrated that *E. coli*-expressed mAMCase is remarkably stable across a broad pH range (*Kashimura et al., 2013*). Similar experiments have not yet been performed on hAMCase. *Olland et al., 2009* previously identified Arg145, His208, and His269 as important for pH specificity . *Seibold et al., 2009* argued that hAMCase isoforms containing asthma protective mutations N45D, D47N, and M61R, which are wildtype in mAMCase, may influence the pKa of Asp138-Glu140 by undergoing structural rearrangement . *Tabata et al., 2022* identified mutations across the course of evolution in Carnivora that were inactivating or structurally destabilizing (loss of S-S bonds; ). *Okawa et al., 2016* identified how primate AMCase lost activity by integration of specific, potentially pKa-shifting, mutations relative to the mouse counterpart .

To this end, we explored sequence differences between mouse and human AMCase homologs for insight into why mAMCase has such high enzymatic activity at pH 2.0 and 6.5 compared to hAMCase. We identified ionizable residues on mAMCase that likely contribute to its overall stability and are not present in hAMCase. Mutations Lys78Gln, Asp82Gly, and Lys160Gln result in the loss of surface-stabilizing salt bridges in hAMCase and may contribute to its reduced activity at more acidic pH. It is likely that the dual pH optima of mAMCase is intrinsic to the catalytic mechanism, where Glu140 can be protonated directly from solution (at low pH) or through proton shuttling across the catalytic triad (at neutral pH; *Figure 1E*). However, hAMCase is likely too destabilized at low pH to observe an increase in $k_{cat}$. hAMCase may be under less pressure to maintain high activity at low pH due to humans' noninsect-based diet, which contains less chitin compared to other mammals with primarily insect-based diets (*Tabata et al., 2022*).

Together, these data demonstrate the importance of using structural and biochemical assays to develop our understanding of the catalytic mechanism governing mAMCase activity. Using biochemical and structural methods, we have developed a detailed model of how AMCase fulfills its role in chitin recognition and degradation. Small chitin oligomers are ideal for measuring the ability of AMCase to cleave β–1,4-glycosidic linkages between GlcNAc units, but these small oligomers do not represent the complex crystalline chitin encountered by AMCase in the lung. It is difficult to extrapolate the effects we observe using small chitin oligomers to binding ($k$on), processivity ($k$proc), catalysis ($k$cat), or product release ($k$off) on the native large and heterogeneous oligomeric substrates. In the future, we hope to be able to directly visualize the mAMCase-chitin interactions and characterize each step of the catalytic mechanism including decrystallization, degradation, product release, and chain sliding (also known as processivity).

To further understand the impact of pH on the structure of AMCase, it will be necessary to crystallize AMCase across a broader pH range that may expose conformational and structural changes that contribute to mAMCase's unique pH activity profile. Our simulations have important limitations that could be overcome by quantum mechanical simulations that allow for changes in protonation state and improved consideration of polarizability. Further, neutron diffraction crystallography could provide novel critical insight into the placement of protons across the active site and help to develop a more complete model of mAMCase's catalytic mechanism at different pH. Understanding the mechanistic basis behind an enzyme's dual pH optima will enable us to engineer proteins with tunable pH optima to develop improved enzyme variants for therapeutic purposes for diseases, such as asthma and lung fibrosis.

## Methods

### Protein expression and purification

Protein expression and purification mAMCase catalytic domain (UniProt: Q91XA9; residues 22–391) was cloned into a pTwist CMV [pmRED006; Twist Biosciences; Addgene ID: 200228] or pcDNA3.1(+) [pmRED013; Genscript; Addgene ID: 200229] expression vector with a C-terminal 6xHis tag. To express mAMCase catalytic domain, 0.8–1 µg/mL plasmid DNA was transfected into ExpiCHO-S cells (ThermoFisher Scientific #A29127) using the Max Titer protocol (ThermoFisher Scientific MAN0014337). After cells were grown shaking at 37 °C with 8% $CO_2$ for 18–22 hours, ExpiFectamine CHO Enhancer (ThermoFisher Scientific #A29129) and ExpiCHO feed (ThermoFisher Scientific #A29129) was added to the flask. Cells were then transferred to 32 °C with 5% $CO_2$ for an additional 9–13 days of growth, with a second volume of ExpiCHO feed added to the flask on day 5 post-transfection. Cells were removed by centrifugation at 4000 RCF for 15 min at 4 °C, and the remaining supernatant was filtered using a 0.22 µm filter at 4 °C. Filtered supernatant was either dialyzed into Ni–nitrilotriacetic acid (NTA) loading buffer [100 mM Tris-HCl (pH 8.0), 150 mM NaCl] at 4 °C in a 10 kDa molecular weight cutoff (MWCO) Slide-A-Lyzer Dialysis Cassette, (ThermoFisher Scientific #66810) for 18–24 hr or concentrated in a 10 kDa MWCO centrifugal concentrator (Amicon #UFC901008) at 4000 RCF in 5 min intervals until the final volume was equal to 10 mL, which was then diluted 1:10 with loading buffer for a total volume of 100 mL. The dialyzed supernatant volume was filtered using a 0.22 µm filter at 4 °C. All purification steps were performed at 4 °C using an ÄKTA fast protein liquid chromatography system (Cytiva). The dialyzed supernatant was applied to a 5 ml HisTrap FF column (Cytiva, 17525501). The column was washed with 40 mL of loading buffer followed by 25 mL of 10% Ni-NTA elution buffer [100 mM Tris-HCl (pH 8.0), 150 mM NaCl, 500 mM imidazole] and then eluted over a 50 mL gradient from 10% to 100% elution buffer. Eluted protein was concentrated to 2.5 mL using a 10 kDa MWCO centrifugal concentrator (Amicon, UFC901024). The sample was further purified by size exclusion chromatography (SEC) using a HiLoad 16/600 Superdex 75 pg column (Cytiva, 28989333) equilibrated with SEC buffer [25 mM Tris-HCl (pH 8.0), 50 mM NaCl]. Eluted fractions were collected and stored at 4 °C for further use.

### 4MU-chitobioside endpoint assay

Chitinase catalytic activity has previously been assayed using 4-methylumbelliferyl chitobioside (4MU-CB; Sigma-Aldrich M9763) (*O'Brien and Colwell, 1987*; *Renkema et al., 1995*). 100 nM chitinase enzyme was incubated with varying concentrations of 4MU-chitobioside up to 117 µM in McIlvaine Buffer at 37 °C (*Barad et al., 2020*). The 4-methylumbelliferone (4MU) fluorophore is quenched by a ß-glycosidic linkage to a short chitin oligomer, which is cleaved by a chitinase enzyme, which generates fluorescence with peak excitation at 360 nm and emission at 450 nm. 4MU fluorescence is pH-dependent with peak excitation at 360 nm and emission at 450 nm at pH 7.0. It has been previously reported that 4MU peak excitation/emission increases and fluorescence intensity decreases as pH becomes more acidic (*Zhi et al., 2013*). Given the pH-dependent fluorescence properties of the 4MU fluorophore, we incubate the reaction at different pH, then quench with 0.1 M Gly-NaOH pH 10.7. Quenching the reaction with 0.1 M Gly-NaOH pH 10.7 stops the enzyme reaction and shifts the pH to maximize the quantum yield of the 4MU substrate.

A Tecan Spark multimode microplate reader is pre-heated to 37 °C. 4MU-chitobioside (Sigma-Aldrich M9763) and AMCase are separately pre-incubated at 37 °C for 15 min. Twenty-five µL of 4MU-chitobioside or McIlvaine Buffer (Boston Bioproducts) is transferred into each well in a Multiplate 96-Well PCR Plate, high profile, unskirted, clear (Bio-Rad MLP9601). Using a Multidrop Combi Reagent Dispenser (Thermo Scientific #5840300), 25 µL of either 100 nM AMCase or McIlvaine Buffer (Boston Bioproducts) is dispensed into each well in the Multiplate 96-Well PCR Plate (Corning #3993). The Multiplate 96-Well PCR Plate is then incubated at 37 °C in a 96-well Non-Skirted PCR Plate Block (Thermo Scientific #88870120) in a digital dry bath (Thermo Scientific #88870006).

The reaction is quenched with 50 µL 0.1 M Gly-NaOH pH 10.7 at timepoints 0", 15", 30", 45", 60", 90". Forty µL of the quenched reaction is transferred to a 384-well Low Volume Black Flat Bottom Polystyrene NBS Microplate (Corning #3820), then immediately read using the following parameters:

- Excitation - 360 nm, 20 nm bandwidth
- Emission - 450 nm, 20 nm bandwidth

- Gain - 50
- Flashes - 20

This assay was performed in quadruplicate for each pH unit reported. This allowed us to reliably measure initial rates of catalysis across a large range of pH conditions. The workflow for this assay is illustrated in (*Figure 1*). A detailed protocol for this assay can be found on (protocols.io).

## Analysis of kinetic data

Twenty-five µL of 200 µM 4MU fluorophore (Sigma-Aldrich M1381) was serially diluted into 25 µL McIl-vaine Buffer (Boston Bioproducts) across the range of pHs to obtain five diluted ligand concentrations ranging from 100 µM to 6.25 µM as well as ligand free. This dilution series was performed in duplicate per 96-Well PCR plate for a total of 8 replicates per ligand concentration at each given pH value. At the end of the experiment, the 4MU dilution series is quenched with 50 µL 0.1 M Gly-NaOH pH 10.7 for a final dilution series ranging from 50 µM to 3.125 µM.

Relative fluorescence (RFU) was plotted against 4MU concentration, then a simple linear regression with the constraint Y=0 when X=0 was performed to obtain a standard curve. We then used the equation Y=mX + b, where m is the slope from the standard curve and Y is the RFU from a given experimental data point, to determine the concentration of 4MU [µM] generated by AMCase at a given time point.

Average 4MU concentration [µM] (n=4) was plotted as a function of time with error bars representing the standard deviation. We then fit a simple linear regression with the constraint Y=0 when X=0 to obtain the initial rate of enzyme activity (4MU [µM]/sec) at each concentration of 4MU-chitobioside [µM]. Average initial rate (n=4) was plotted as a function of 4MU-chitobioside concentration [µM] with error bars representing the standard deviation. We fit our data to a Michaelis-Menten function without substrate inhibition to obtain $V_{max}$ and $K_M$ parameters. We used the equation $k_{cat} = V_{max}/$[Enzyme] where [Enzyme]=0.1 µM to calculate $k_{cat}$. We calculated catalytic efficiency (CE) using the equation CE = $K_M/k_{cat}$. Kinetic parameters Vmax, $K_M$, $k_{cat}$, and catalytic efficiency were plotted as a function of pH.

## Apo crystallization

Using hanging-drop vapor diffusion, crystallization screens were performed using a 96-well Clear Flat Bottom Polystyrene High Binding microplate (Corning CLS9018BC) with 0.5 mL of reservoir solution in each well. Crystallization drops were set up on 96-well plate seals (SPT Labtech 4150–05100) with 0.2 µl of AMCase at 11 mg/ml and 0.2 µl of reservoir using an SPT Labtech mosquito crystal. After 21 days at 20 °C, we observed crystals in a reservoir solution containing 20% PEG-6000, 0.1 M Sodium Acetate pH 5.0, and 0.2 M Magnesium Chloride (II) (MgCl2) (NeXtal PACT Suite Well A10; #130718).

## Apo data collection, processing, and refinement at cryogenic temperature

Diffraction data were collected at the beamline ALS 8.3.1 at 100 K. Diffraction data from multiple crystals were merged using xia2 (*Winter, 2010*), implementing DIALS (*Winter et al., 2018*) for indexing and integration, and Aimless (*Winn et al., 2011*) for scaling and merging. We confirmed the space group assignment using DIMPLE (*Wojdyr et al., 2013*). We calculated phases by the method of molecular replacement, using the program Phaser (*McCoy et al., 2007*) and a previous structure of hAMCase (PDB: 3FXY) as the search model. The model was manually adjusted in Coot to fit the electron density map calculated from molecular replacement, followed by automated refinement of coordinates, atomic displacement parameters, and occupancies using phenix.refine (*Afonine et al., 2012*) with optimization of restraint weights. Default refinement parameters were used, except the fact that five refinement macrocycles were carried out per iteration and water molecules were automatically added to peaks in the 2mFo-DFc electron density map higher than 3.5 Å. The minimum model-water distance was set to 1.8 Å, and a maximum model-water distance was set to 6 Å. For later rounds of refinement, hydrogens were added to riding positions using *phenix.ready_set*, and B-factors were refined anisotropically for non-hydrogen and non-water atoms. Following two initial rounds of iterative model building and refinement using the aforementioned strategy, we began introducing additional parameters into the model, enabled by the extraordinarily high resolution of our diffraction data. First, we implemented anisotropic atomic displacement parameters for heavy atoms (C, N, O,

and S), followed by refinement of explicit hydrogen atom positions. A final round of refinement was performed without updating water molecules.

## Apo data collection, processing, and refinement at room temperature
Diffraction data were collected at the beamline ALS 8.3.1 at 277 K. Data collection, processing, refinement, and model building were performed as described previously for the apo crystals at cryogenic temperature.

## Holo crystallization
Initially, crystals were grown by hanging-drop vapor diffusion with a reservoir solution containing 20% PEG-6000 (Hampton Research HR2533), 0.1 M Sodium Acetate (pH 3.6, Hampton Research HR293301; pH 4.1, Hampton Research HR293306; pH 5.0, Hampton Research HR293315; pH 5.6, Hampton Research HR293321), and 0.2 M Magnesium Chloride (II) (MgCl2) (Hampton Research HR2559). Screens were performed using a 96-well Clear Flat Bottom Polystyrene High Binding microplate (Corning CLS9018BC) with 0.5 mL of reservoir solution in each well. Crystallization drops were set up on 96-well plate seals (SPT Labtech 4150–05100) with 0.2 µl of AMCase at 11 mg/ml and 0.2 µl of reservoir using an SPT Labtech mosquito crystal. Crystals grew after 1–2 days at 20 °C.

Using hanging drop diffusion vapor, holo crystals grew after 12 hours at 20 °C. For the holo form with $GlcNAc_2$ (Megazyme O-CHI2), this construct crystallized in either $P2_12_12$ or $P2_12_12_1$ with either 2 or 4 molecules in the ASU and diffracted to a maximum resolution between 1.50–1.95 Å. For the holo form with $GlcNAc_3$ (Megazyme O-CHI3), this construct crystallized in $P2_12_12$ with 2 molecules in the ASU and diffracted to a maximum resolution of 1.70 Å.

## Holo data collection, processing, and refinement at cryogenic temperature
Diffraction data were collected at the beamline ALS 8.3.1 and SSRL beamline 12–1 at 100 K. Data collection, processing, refinement, and model building were performed as described previously for the apo crystals.

Ligands were modeled into 2mFo-DFc maps with Coot, using restraints generated by *phenix.elbow* from an isomeric SMILES (simplified molecular input line-entry system) string (*Emsley and Cowtan, 2004*) using AM1 geometry optimization. Default refinement parameters were used, except the fact that five refinement macrocycles were carried out per iteration and water molecules were automatically added to peaks in the 2mFo-DFc electron density map higher than 3.5 Å. The minimum model-water distance was set to 1.8 Å, and a maximum model-water distance was set to 6 Å. Changes in protein conformation and solvation were also modeled. Hydrogens were added with *phenix.ready_set*, and waters were updated automatically. A final round of refinement was performed without updating water molecules (*Wojdyr et al., 2013*).

## Ligand modeling
For consistency, ligands were assigned an alternative conformation ID based on the sugar-binding subsites it occupied:

> $GlcNAc_2$ ResID 401 Conf. A, –3 to –2
> $GlcNAc_2$ ResID 401 Conf. B, –2.5 to –1.5
> $GlcNAc_2$ ResID 401 Conf. C, –2 to –1
> $GlcNAc_2$ ResID 402 Conf. D, –1 to +1
> $GlcNAc_2$ ResID 402 Conf. B, +1 to+2
> $GlcNAc_2$ ResID 402 Conf. A, +1' to +2
> $GlcNAc_3$ ResID 401 Conf. A, –4 to –2
> $GlcNAc_3$ ResID 401 Conf. B, –3 to –1
> $GlcNAc_3$ ResID 401 Conf. C, –2 to +1
> $GlcNAc_3$ ResID 402 Conf. B, –1 to +2

Ligand occupancies and B-factors using *phenix.refine*. Ligands with occupancies ≤0.10 were removed from the model.

## Ringer analysis

Individual residues in each of the mAMCase structures were run through Ringer using mmtbx.ringer. Outputs from the csv file were then plotted using Matplotlib.

## pKa analysis

We used the APBS-PDB2PQR software suite (https://server.poissonboltzmann.org/pdb2pqr; *Jurrus et al., 2018*). Each PDB model was separated into two separate models containing a single Asp138 conformation in either the *inactive* (down towards Asp136) or *active conformation* (up towards Glu140). Solvent and ligand molecules were not modified. The pH of the crystallization condition was provided for PROPKA to assign protonation states. The default forcefield PARSE was used. The following additional options were selected: Ensure that new atoms are not rebuilt too close to existing atoms; Optimize the hydrogen bonding network.

## Molecular dynamics

Simulations were performed using hexaacetyl-chitohexaose (PubChem Compound ID: 6918014) modeled into 8GCA with Asp138 in either the *inactive* (down towards Asp136) or *active conformation* (up towards Glu140). The model PDB file was opened in MOE and solvated in a sphere of water 10 Å away from the protein. This system then underwent structural preparation for simulations using the standard parameters with the AMBER14 forcefield. The system then was protonated to set pH {2.0, 6.5} based on sidechain pKa predictions using the 3DProtonate menu followed by confirmation of appropriate protonation by PROPKA calculations. Protonated models underwent energy minimization by steepest descent before simulations were set up. Equilibration was performed for 10 ps followed by 100 ps of thermal gradient equilibration from 0K to 300K. A thermal bath equilibration was run for 100 ps before the production runs were started. Productions were run for 10 ns with a time step of 0.5 fs to not overshoot bond vibrations. The simulation was sampled every 10 ps for subsequent data analysis which was performed using the MOE database viewer and replotted using GraphPad Prism.

## Acknowledgements

We are grateful to Aashish Manglik and Mingliang Jin for providing ExpiCHO-S cells; to Liam McKay and Jose Luis Olmos, Jr. for support with the X-ray crystallography facility at UCSF; to George Meigs for assistance with X-ray data collection at ALS 8.3.1.; to Tzanko Doukov for assistance with X-ray data collection at SSRL 12–2; to Eric Greene, Duncan Muir, Stephanie Wankowicz, and Benjamin Barad for helpful discussions and critical feedback. Structural biology applications used at UCSF were compiled and configured by SBGrid (*Morin et al., 2013*).

This work was supported, in part, by California's Tobacco Related Disease Research Program (TRDRP) grant T29IP0554 (J.S.F). Research reported in this publication was supported by the National Heart, Lung, and Blood Institute of the National Institutes of Health under award number R01HL148033 (S.J.V.D., J.S.F.). Beamline 8.3.1 at the Advanced Light Source is operated by the University of California Office of the President, Multicampus Research Programs and Initiatives grant MR-15–328599, NIH (R01 GM124149 and P30 GM124169), Plexxikon Inc, and the Integrated Diffraction Analysis Technologies program of the US Department of Energy Office of Biological and Environmental Research. The crystallographic data was collected using beamlines at the Advanced Light Source, and the Stanford Synchrotron Radiation Lightsource. The Advanced Light Source (Berkeley, CA) is a national user facility operated by Lawrence Berkeley National Laboratory on behalf of the US Department of Energy under contract number DE-AC02-05CH11231, Office of Basic Energy Sciences. Use of the Stanford Synchrotron Radiation Lightsource, SLAC National Accelerator Laboratory, is supported by the U.S. Department of Energy, Office of Science, Office of Basic Energy Sciences under Contract No. DE-AC02-76SF00515. The SSRL Structural Molecular Biology Program is supported by the DOE Office of Biological and Environmental Research, and by the National Institutes of Health, National Institute of General Medical Sciences (including P41GM103393). This material is based upon work supported by the National Science Foundation Graduate Research Fellowship Program under Grant No. (1650113; R.E.D.). Any opinions, findings, and conclusions or recommendations expressed in this material are those of the author(s) and do not necessarily reflect the views of the National Science

Foundation. R.E.D. is a Howard Hughes Medical Institute Gilliam Fellow. R.M.L. is supported by the Howard Hughes Medical Institute.

## Additional information

### Competing interests

Steven Van Dyken, Richard M Locksley: S.J.V.D. and R.M.L. are listed as inventors on a patent United States application: 17/505,561 for the use of chitinases to treat fibrotic lung disease. S.J.V.D., R.M.L., and J.S.F. are listed as inventors on a patent for mutant chitinases with enhanced expression and activity. James S Fraser: S.J.V.D. and R.M.L. are listed as inventors on a patent for the use of chitinases to treat fibrotic lung disease. United States application: 17/505,561 S.J.V.D., R.M.L., and J.S.F. are listed as inventors on a patent for mutant chitinases with enhanced expression and activity. The other authors declare that no competing interests exist.

### Funding

| Funder | Grant reference number | Author |
| --- | --- | --- |
| University of California | T29IP0554 | James S Fraser |
| National Heart, Lung, and Blood Institute | R01HL148033 | Steven Van Dyken James S Fraser |
| National Institute of General Medical Sciences | P30GM124169 | James S Fraser |
| National Science Foundation Graduate Research Fellowship Program | 1650113 | Roberto Efraín Díaz |
| Howard Hughes Medical Institute | | Roberto Efraín Díaz Richard M Locksley |

The funders had no role in study design, data collection and interpretation, or the decision to submit the work for publication.

### Author contributions

Roberto Efraín Díaz, Conceptualization, Data curation, Software, Formal analysis, Investigation, Visualization, Methodology, Writing – original draft, Writing – review and editing; Andrew K Ecker, Formal analysis, Investigation, Visualization, Methodology, Writing – original draft; Galen J Correy, Pooja Asthana, Michael C Thompson, Investigation, Visualization; Iris D Young, Bryan Faust, Investigation; Ian B Seiple, Resources, Supervision, Investigation; Steven Van Dyken, Conceptualization, Supervision, Funding acquisition, Investigation, Writing – review and editing; Richard M Locksley, Conceptualization, Supervision, Funding acquisition, Writing – review and editing; James S Fraser, Conceptualization, Resources, Supervision, Funding acquisition, Visualization, Methodology, Writing – original draft, Project administration, Writing – review and editing

### Author ORCIDs

Roberto Efraín Díaz http://orcid.org/0000-0002-1172-9919
James S Fraser https://orcid.org/0000-0002-5080-2859

Reviewer #1 (Public Review): https://doi.org/10.7554/eLife.89918.3.sa1
Author response https://doi.org/10.7554/eLife.89918.3.sa2

## Additional files

### Supplementary files

• Supplementary file 1. Occupancy of each ligand subsite and Asp138 in the active conformation (separate file).

- Supplementary file 2. pKa across Asp136, Asp138, Glu140 of mAMCase structures in either Asp138 inactive or Asp138 active conformation (separate file).
- MDAR checklist

## Data availability

Structural data are available in the PDB, PDB accession numbers are provide in Table 1 and referred to in the article text and figure legends. All other data needed to reproduce figures are deposited in Zenodo (https://doi.org/10.5281/zenodo.8250616).

The following datasets were generated:

| Author(s) | Year | Dataset title | Dataset URL | Database and Identifier |
|---|---|---|---|---|
| Díaz RE, Fraser JS | 2023 | Kinetic properties of mAMCase catalytic domain at various pH | https://doi.org/10.5281/zenodo.8250616 | Zenodo, 10.5281/zenodo.8250616 |
| Diaz RE, Correy GJ, Young ID, Thompson MC, Fraser JS | 2022 | Apo mouse acidic mammalian chitinase, catalytic domain at 100 K | https://www.rcsb.org/structure/8FG5 | RCSB Protein Data Bank, 8FG5 |
| Diaz RE, Asthana P, Fraser JS | 2022 | Apo mouse acidic mammalian chitinase, catalytic domain at 277 K | https://www.rcsb.org/structure/8FG7 | RCSB Protein Data Bank, 8FG7 |
| Diaz RE, Fraser JS | 2024 | Mouse acidic mammalian chitinase, catalytic domain in complex with N,N',N''-triacetylchitotriose at pH 4.74 | https://www.rcsb.org/structure/8GCA | RCSB Protein Data Bank, 8GCA |
| Diaz RE, Fraser JS | 2023 | Mouse acidic mammalian chitinase, catalytic domain in complex with N,N'-diacetylchitobiose at pH 4.91 | https://www.rcsb.org/structure/8FRC | RCSB Protein Data Bank, 8FRC |
| Diaz RE, Fraser JS | 2023 | Mouse acidic mammalian chitinase, catalytic domain in complex with N,N'-diacetylchitobiose at pH 5.08 | https://www.rcsb.org/structure/8FR9 | RCSB Protein Data Bank, 8FR9 |
| Diaz RE, Fraser JS | 2023 | Mouse acidic mammalian chitinase, catalytic domain in complex with N,N'-diacetylchitobiose at pH 5.25 | https://www.rcsb.org/structure/8FRB | RCSB Protein Data Bank, 8FRB |
| Diaz RE, Fraser JS | 2023 | Mouse acidic mammalian chitinase, catalytic domain in complex with N,N'-diacetylchitobiose at pH 5.25 | https://www.rcsb.org/structure/8FRD | RCSB Protein Data Bank, 8FRD |
| Diaz RE, Fraser JS | 2023 | Mouse acidic mammalian chitinase, catalytic domain in complex with N,N'-diacetylchitobiose at pH 5.43 | https://www.rcsb.org/structure/8FRG | RCSB Protein Data Bank, 8FRG |
| Diaz RE, Fraser JS | 2023 | Mouse acidic mammalian chitinase, catalytic domain in complex with diacetylchitobiose at pH 5.60 | https://www.rcsb.org/structure/8FRA | RCSB Protein Data Bank, 8FRA |

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
