## [Editor Report · eLife assessment]

This structural and biochemical study of the mouse homolog of acidic mammalian chitinase (AMCase) enhances our understanding of the pH-dependent activity and catalytic properties of mouse AMCase, and it sheds light on its adaptation to different physiological pH environments. The methods and analysis of data are **solid**, providing several lines of evidence to support the development of mechanistic hypotheses. While the findings and interpretation will be **valuable** to those studying AMCase in mice, the broader significance, including extension of the results to other species including human, remain less clear.

---

## [Referee Report · Reviewer #1 (Public Review)]

General comments:

This paper investigates the pH-specific enzymatic activity of mouse acidic mammalian chitinase (AMCase) and aims to elucidate its function's underlying mechanisms. The authors employ a comprehensive approach, including hydrolysis assays, X-ray crystallography, theoretical calculations of pKa values, and molecular dynamics simulations to observe the behavior of mouse AMCase and explore the structural features influencing its pH-dependent activity.

The study's key findings include determining kinetic parameters (Kcat and Km) under a broad range of pH conditions, spanning from strong acid to neutral. The results reveal pH-dependent changes in enzymatic activity, suggesting that mouse AMCase employs different mechanisms for protonation of the catalytic glutamic acid residue and the neighboring two aspartic acids at the catalytic motif under distinct pH conditions.

The novelty of this research lies in the observation of structural rearrangements and the identification of pH-dependent mechanisms in mouse AMCase, offering a unique perspective on its enzymatic activity compared to other enzymes. By investigating the distinct protonation mechanisms and their relationship to pH, the authors reveal the adaptive nature of mouse AMCase, highlighting its ability to adjust its catalytic behavior in response to varying pH conditions. These insights contribute to our understanding of the pH-specific enzymatic activity of mouse AMCase and provide valuable information about its adaptation to different physiological conditions.

Overall, the study enhances our understanding of the pH-dependent activity and catalytic properties of mouse AMCase and sheds light on its adaptation to different physiological pH environments.

Comments on revised version:

In their revised manuscript, the authors have made significant efforts to address the reviewers' comments.

---

## [Author Response]

The following is the authors’ response to the original reviews.

**eLife assessment**
This structural and biochemical study of the mouse homolog of acidic mammalian chitinase (AMCase) enhances our understanding of the pH-dependent activity and catalytic properties of mouse AMCase and sheds light on its adaptation to different physiological pH environments. The methods and analysis of data are solid, providing several lines of evidence to support a development of mechanistic hypotheses. While the findings and interpretation will be valuable to those studying AMCase in mice, the broader significance, including extension of the results to other species including human, remain unclear.
**Public Reviews:**

**Reviewer #1 (Public Review):**
General comments:This paper investigates the pH-specific enzymatic activity of mouse acidic mammalian chitinase (AMCase) and aims to elucidate its function's underlying mechanisms. The authors employ a comprehensive approach, including hydrolysis assays, X-ray crystallography, theoretical calculations of pKa values, and molecular dynamics simulations to observe the behavior of mouse AMCase and explore the structural features influencing its pH-dependent activity.The study's key findings include determining kinetic parameters (Kcat and Km) under a broad range of pH conditions, spanning from strong acid to neutral. The results reveal pH-dependent changes in enzymatic activity, suggesting that mouse AMCase employs different mechanisms for protonation of the catalytic glutamic acid residue and the neighboring two aspartic acids at the catalytic motif under distinct pH conditions.The novelty of this research lies in the observation of structural rearrangements and the identification of pH-dependent mechanisms in mouse AMCase, offering a unique perspective on its enzymatic activity compared to other enzymes. By investigating the distinct protonation mechanisms and their relationship to pH, the authors reveal the adaptive nature of mouse AMCase, highlighting its ability to adjust its catalytic behavior in response to varying pH conditions. These insights contribute to our understanding of the pH-specific enzymatic activity of mouse AMCase and provide valuable information about its adaptation to different physiological conditions.Overall, the study enhances our understanding of the pH-dependent activity and catalytic properties of mouse AMCase and sheds light on its adaptation to different physiological pH environments.
**Reviewer #2 (Public Review):**
Summary:In this study of the mouse homolog of acidic mammalian chitinase, the overall goal is to provide a mechanistic explanation for the unusual observation of two pH optima for the enzyme. The study includes biochemical assays to establish kinetic parameters at different solution pH, structural studies of enzyme/substrate complexes, and theoretical analysis of amino acid side chain pKas and molecular dynamics.Strengths:The biochemical assays are rigorous and nicely complemented by the structural and computational analysis. The mechanistic proposal that results from the study is well rationalized by the observations in the study.Weaknesses:The overall significance of the work could be made more clear. Additional details could be provided about the limitations of prior biochemical studies of mAMC that warranted the kinetic analysis. The mouse enzyme seems unique in terms of its behavior at high and low pH, so it remains unclear how the work will enhance broader understanding of this enzyme class. It was also not clear can the findings be used for therapeutic purposes, as detailed in the abstract, if the human enzyme works differently.

We have edited the paper to address these concerns

**Recommendations for the authors:**

**Reviewer #1 (Recommendations For The Authors):**
Major comments:(1) Regarding the pH profiles of mouse AMCase, previous studies have reported its activity at pH 2.0 and within the pH range of 3-7. In this paper, the authors conducted kinetic measurements and showed that pH 6.5 is optimal for kcat/Km. The authors emphasize the significance of mouse AMCase's activity in the neutral region, particularly at pH 6.5, for understanding its physiological relevance in humans. To provide a comprehensive overview, it would be valuable for the authors to summarize the findings from previous and current studies, discuss their implications for future pulmonary therapy in humans, and cite relevant literature. Additionally, the authors should highlight their research's specific contributions and novel findings, such as the determination of kinetic parameters (Kcat and Km) under different pH conditions. Emphasizing why previous studies may have required these observations and underscoring the importance of the present findings in addressing those knowledge gaps will help readers understand the significance of the study and its impact on the field of enzymology.

We thank the reviewer for this comment. In keeping with the knowledge gaps addressed directly by this paper, we have not augmented the discussion of future pulmonary therapy in humans. We have summarized the present findings at the end of the introduction as follows:

“We measured the mAMCase hydrolysis of chitin, which revealed significant activity increase under more acidic conditions compared to neutral or basic conditions. To understand the relationship between catalytic residue protonation state and pH-dependent enzyme activity, we calculated the theoretical pKa of the active site residues and performed molecular dynamics (MD) simulations of mAMCase at various pHs. We also directly observed conformational and chemical features of mAMCase between pH 4.74 to 5.60 by solving X-ray crystal structures of mAMCase in complex with oligomeric GlcNAcn across this range.”

(2) Regarding the implications of the pKa values and Asp138 orientation for the pH optima, it would be valuable for the authors to discuss the variations in optimal activity by pH among GH-18 chitinases and investigate the underlying factors contributing to these differences. In particular, exploring the role of Asp138 orientation in chitotriosidase, another mammalian chitinase, would provide important insights. Chitotriosidase is known to be inactive at pH 2.0, and it would be interesting to investigate whether the observed orientation of Asp138 towards Glu140 in mouse AMCase for pH 2.0 activity is lacking in chitotriosidase.

There are similar rotations of the two acidic residues in the literature on Chit1. The variety of crystal pH conditions and the lack of a straightforward mechanism for pKa shifts in AMCase make it difficult to draw a comparison to why Chit1 is inactive at low pH, but this is an interesting area for future study. See a more full discussion in: https://www.ncbi.nlm.nih.gov/pmc/articles/PMC2760363/

Furthermore, considering the lower activity of human AMCase at pH 2.0, it would be worthwhile to examine whether the Asp138 orientation towards Glu140, as observed in mouse AMCase, is also absent in human AMCase. Exploring this aspect will help determine if the orientation of Asp138 plays a critical role in pH-dependent activity in human AMCase.

The situation for hAMCase is similar to Chit1 as the rotations observed here for mAMCase are also present. It is not the whether Asp138 can rotate, but rather the relevant energetic penalties as we discuss in the manuscript.

(3) In a previous study by Okawa et al.(Loss and gain of human acidic mammalian chitinase activity by nonsynonymous SNPs. Mol Biol Evol 33, 3183-3193, 2016), it was reported that specific amino acid substitutions (N45D, D47N, and R61M) encoded by nonsynonymous single nucleotide polymorphisms (nsSNPs) in the N-terminal region of human AMCase had distinct effects on its chitinolytic activity. Introducing these three residues (N45D, D47N, and R61M) could activate human AMCase. This activation significantly shifted the optimal pH from 4-5 to 2.0.Considering the significant impact of these amino acid substitutions on the pH-dependent activity of human AMCase, the authors should discuss this point in the manuscript's discussion section. Incorporating the findings and relating them to the current study's observations on pH optima and Asp138 orientation can provide a comprehensive understanding of the factors influencing pH-dependent activity in AMCase.

We added a citation and dicuss how the mutations identified by this study could potentially shift the pKa of key catalytic residues:

“Okawa et al identified how primate AMCase lost activity by integration of specific, potentially pKa-shifting, mutations relative to the mouse counterpart42b.”

(4) To further strengthen the discussion, the authors could explore the ancestral insectivorous nature of placental mammals and the differences in chitinase activity between herbivorous and omnivorous species. Incorporating these aspects would add depth and relevance to the overall discussion of AMCase. AMCase is an enzyme known for its role in digesting insect chitin in the stomachs of various insectivorous and omnivorous animals, including bats, mice, chickens, pigs, pangolins, common marmosets, and crab-eating monkeys 1-7. However, in certain animals, such as dogs (carnivores) and cattle (herbivores), AMCase expression and activity are significantly low, leading to impaired chitin digestion 8. These observations suggest a connection between dietary habits and the expression and activity of the AMCase gene, ultimately influencing chitin digestibility across different animal species 8.(1) Strobelet al. (2013). Insectivorous bats digest chitin in the stomach using acidic mammalian chitinase. PloS one 8, e72770.(2) Ohno et al. (2016). Acidic mammalian chitinase is a proteases-resistant glycosidase in mouse digestive system. Sci Rep 6, 37756.(3) Tabata et al. (2017). Gastric and intestinal proteases resistance of chicken acidic chitinase nominates chitin-containing organisms for alternative whole edible diets for poultry. Sci Rep 7, 6662.(4) Tabata et al. (2017). Protease resistance of porcine acidic mammalian chitinase under gastrointestinal conditions implies that chitin-containing organisms can be sustainable dietary resources. Sci Rep 7, 12963.(5) Ma et al. (2018). Acidic mammalian chitinase gene is highly expressed in the special oxyntic glands of Manis javanica. FEBS Open Bio 8, 1247-1255.(6) Tabata et al. (2019). High expression of acidic chitinase and chitin digestibility in the stomach of common marmoset (Callithrix jacchus), an insectivorous nonhuman primate. Sci. Rep. 9. 159.(7) Uehara et al. (2021). Robust chitinolytic activity of crab-eating monkey (Macaca fascicularis) acidic chitinase under a broad pH and temperature range. Sci. Rep. 11, 15470.(8) Tabata et al. (2018). Chitin digestibility is dependent on feeding behaviors, which determine acidic chitinase mRNA levels in mammalian and poultry stomachs. Sci Rep 8, 1461.

This overall point is covered by our brief discussion on diet differences:

“However, hAMCase is likely too destabilized at low pH to observe an increase in _k_cat. hAMCase may be under less pressure to maintain high activity at low pH due to humans’ noninsect-based diet, which contains less chitin compared to other mammals with primarily insect-based diets42. “

(5) It is important for the authors to clearly state the limitations of their simulations and emphasize the need for experimental validation or additional supporting evidence. This will provide transparency and enable readers to understand the boundaries of the study's findings. A comprehensive discussion of limitations would contribute to a more robust interpretation of the results.

We added a sentence to the discussion:

“Our simulations have important limitations that could be overcome by quantum mechanical simulations that allow for changes in protonation state and improved consideration of polarizability.”

Minor comments:(1) Regarding the naming of AMCase, it is important to accurately describe it based on its acidic isoelectric point rather than its enzymatic activity under acidic conditions based on the original paper (Reference #14 (Boot, R. G. et al). Identification of a novel acidic mammalian chitinase distinct from chitotriosidase. J. Biol. Chem. 276, 6770-6778 (2001)).

We have made this modification

(2) In the introduction, providing more context regarding the terminology of acidic mammalian chitinase (AMCase) would be beneficial. While AMCase was initially discovered in mice and humans, subsequent research has revealed its presence in various vertebrates, including birds, fish, and other species. Therefore, it would be appropriate to include the alternative enzyme name, Chia (chitinase, acidic), in the introduction to reflect its broader distribution across different organisms. This clarification would enhance the readers' understanding of the enzyme's taxonomy and facilitate further exploration of its functional significance in diverse biological systems.

We have made this modification

(3) The authors mention that AMCase is active in tissues with neutral pHs, such as the lung. However, it is important to consider that the pH in the lung is lower, around 5, due to the presence of dissolved CO2 that forms carbonic acid. The lung microenvironment is known to vary, and specific regions or conditions within the lung may have slightly different pH levels. By addressing the pH conditions in the lungs and their relationship to AMCase's activity, the authors can enhance our understanding of the enzyme's function within its physiological context. A thorough discussion of the specific pH conditions in the lung and their implications for AMCase's activity would provide valuable insights into the enzyme's role in lung pathophysiology.

To keep the focus on the insights we have made, we have elected not to expand this discussion.

(4) It would be helpful for the authors to provide more information about the substrate or products of AMCase. The basic X-ray crystal structures used in this study are GlcNAc2 or GlcNAc3, known products of AMCase. Including details about the specific ligands involved in the enzymatic reactions would enhance the understanding of the study's focus.

We are unclear about what this means - and since it is a minor comment, we have elected not to change the discussion of substrates here.

(5) The authors should critically evaluate the inclusion of the term "chitin-binding" in the Abstract and Introduction. Suppose substantial evidence or discussion regarding the specific chitin-binding properties of the enzyme or its relevance to the immune response needs to be included. In that case, removing or modifying that statement might be appropriate.

We are unclear about what this means - and since it is a minor comment, we have elected not to change the discussion of “chitin-binding” here.

(6) The authors developed an endpoint assay to measure the activity of mouse AMCase across a broad pH range, allowing for direct measurement of kinetic parameters. The authors should provide a more detailed description of the methods used, including any specific modifications made to the previous assay, to ensure reproducibility and facilitate further research in the field. It is important to clearly show the novelty of their endpoint assay compared to previous methods employed in other reports. The authors should also explain how their modified endpoint assay differs from existing assays and highlight its advancements or improvements. This will help readers understand the unique features and contributions of the assay in the context of previous methods.

We have included a detailed method description and figures already. See also our previous paper by Barad which includes other, related, assays.

(7) The authors suggest that mouse AMCase may be subject to product inhibition, potentially due to its transglycosylation activity, which can affect the Michaelis-Menten model predictions at high substrate concentrations. However, the reviewer needed help understanding the specific impact of transglycosylation on the kinetic parameters. It would be helpful for the authors to provide a more appropriate and detailed explanation, clarifying how transglycosylation activity influences the kinetic behavior of AMCase and its implications for the observed results.

The experiments to conclusively demonstrate this are beyond our current capabilities.

(8) In the Abstract, the authors state, "We also solved high resolution crystal structures of mAMCase in complex with chitin, where we identified extensive conformational ligand heterogeneity." This reviewer suggests replacing "chitin" with "oligomeric GlcNAcn" throughout the text, specifically about biochemical experiments. It is important to accurately describe the experimental conditions and ligands used in the study.

We have made these changes throughout the manuscript

(9) In the introduction, the authors mention "a polymer of β(1-4)-linked N-acetyl-D-glucosamine (GlcNAc)". In this case, the letter "N" should be italicized to conform to the proper notation for the monosaccharide abbreviation.

corrected (and hopefully would have been done so by the copy editor!)

(10) In the introduction, the authors state, "In the absence of AMCase, chitin accumulates in the airways, leading to epithelial stress, chronic activation of type 2 immunity, and age-related pulmonary fibrosis5,6". It is recommended to clarify that "AMCase" refers to "acidic mammalian chitinase (AMCase)" in this context, as it is the first mention of the enzyme in the introduction.

We moved that section so that it flows better and is introduced with the full name.

(11) In the introduction, the authors state, "Mitigating the negative effects of high chitin levels is particularly important for mammalian lung and gastrointestinal health." This reviewer requests further clarification on the connection between chitin and gastrointestinal health. Please provide an explanation or reference to support this statement.

We have modified this sentence to:

“Chitin levels can be potentially important for mammalian lung and gastrointestinal health.”

(12) In the introduction, the authors mention that "Acidic Mammalian Chitinase (AMCase) was originally discovered in the stomach and named for its high enzymatic activity under acidic conditions." It is recommended to include Reference #14 (Boot et al. J. Biol. Chem. 276, 6770-6778, 2001) as it provides the first report on mouse and human AMCase, contributing to the understanding of the enzyme.However, it is worth noting that while this paragraph primarily focuses on human tissues, Reference #14 primarily discusses mouse AMCase but also reports on human AMCase. Additionally, References #8 and #9 mainly discuss mouse AMCase. This creates confusion in the description of human and mouse AMCase within the paragraph.Considering that this paper aims to focus on the unique features of mouse AMCase, it is suggested that the authors provide a more specific and balanced description of both human and mouse AMCase throughout the main text..

We have clarified the origin of the name AMCase and the results distinguish the two orthologs in the text with h or mAMCase.

(13) Figure 1A in the Introduction section has been previously presented in several papers. The authors should consider moving this figure to the Results section and present an alternative figure based on their experimental results to enhance the novelty and impact of the study.

We have considered this option, but prefer the original placement.

(14) In the Results section, the authors mentioned, "Prior studies have focused on relative mAMCase activity at different pH18,20, limiting the ability to define its enzymological properties precisely and quantitatively across conditions of interest." It would be beneficial for the authors to include reference #14, the first report showing the pH profile of mouse AMCase, to support their statement.

We have added this reference

(15) Regarding the statement, "To overcome the pH-dependent fluorescent properties of 4MU-chitobioside, we reverted the assay into an endpoint assay, which allowed us to measure substrate breakdown across different pH (Supplemental Figure 1A)", the authors should provide a more detailed description of the improvements made to measure AMCase activity. Additionally, it would be helpful to include a thorough explanation of the figure legend for Supplementary Figure 1A to provide clarity to readers.

We have included a detailed method description and figures already. See also our previous paper by Barad which includes other, related, assays.

(16) Figure 1B shows that the authors used the AMCase catalytic domain. It would benefit the authors to explain the rationale behind this choice in the figure legend or the main text.

This point is addressed in the text:

“Previous structural studies on AMCase have focused on interactions between inhibitors like methylallosamidin and the catalytic domain of the protein.”

(17) For Figures 1C-E, it is recommended that the authors include error bars in their results to represent the variability or uncertainty of the data. In Figure 1E, the authors should clarify the units of the Y-axis (e.g., sec-1 µM-1). Additionally, in Figure 1F, the authors should explain how the catalytic acidity is shown.

We have added error bars and axis labels. Figure 1F is conceptual, so we are leaving it as is.

(18) The authors stated, "These observations raise the possibility that mAMCase, unlike other AMCase homologs, may have evolved an unusual mechanism to accommodate multiple physiological conditions." It would be helpful for the authors to compare and discuss the pH-dependent AMCase activity of mouse AMCase with other AMCase homologs to support this statement.

That is an excellent idea for future comparative studies, but beyond the scope of what we are examining in this paper.

(19) The authors should explain Supplemental Figures 1B and C in the Results or Methods sections to provide context for these figures.

We are unclear about what this means - and since it is a minor comment, we have elected not to change these sections.

(20) Supplemental Figure 3 is missing any description. It would be important for the authors to include a mention of this figure in the main text before Supplemental Figure 4 to guide the readers.

The full legend is in there now and the reference to Supplemental 4 was mislabeled.

(21) For Supplemental Figure 4, the authors should explain the shape of the symbol used in the figure. Additionally, they should explain "apo" and "holoenzyme" in the context of this figure.

Unclear what a shape means in this context - perhaps the confusion arises because these are violin plots showing distributions.

(22) Table 1 requires a more detailed explanation of its contents. Additionally, Tables 2 and 3 need to be included. The authors should include these missing tables in the revised version and explain their contents appropriately.

Table 1 is the standard crystallographic table - there isn’t much more detailed explanation that can be offered. Tables 2 and 3 were not transferred properly by BioRxiv but were included in the review packet as requested a day after submission.

(23) In Figure 4, it would be beneficial to enlarge Panels A-C to improve the ease of comprehension for readers. Additionally, it is recommended to use D136, D138, and E140 instead of D1, D2, and E to label the respective parts. The authors should also explain the meaning of the symbol used in the figure.

Since it is a minor comment, we have elected not to change these figures.

(24) In Figure 5, it would be beneficial to enlarge Panels A-C to improve the ease of comprehension for readers.

Since it is a minor comment, we have elected not to change these figures.

(25) Similarly, in Figure 6, all panels should be enlarged to enhance the ease of comprehension for readers.

Since it is a minor comment, we have elected not to change these figures.

**Reviewer #2 (Recommendations For The Authors):**
In general, I did not identify many detailed or technical concerns with the work. A few items for the authors to consider are listed below.(1) The interpretation of the crystallographic datasets seems complicated by the heterogeneity in the substrate component. It might be nice to see more critical analysis of the approach here. Are there other explanations or possible models that were considered? Do other structures of chitinases or other polysaccharide hydrolases exhibit the same phenomenon?

We have tried in writing it to provide a very critical approach to this and it is quite likely that other structures contain unmodeled density containing similar heterogeneity (but it is just unmodeled).

(2) It would be ideal to include more experimental validation of the proposed mechanism. Much of the manuscript includes theoretical validations (pKa estimation, dynamics, etc) - but it would be optimal to make an enzyme variant or do an experiment with a substrate analog.

Yes - we agree that follow on experiments are needed to fully test the mechanism and that those will be the subject of future work.

(3) For an uninitiated reviewer, I think the major issue with this study is that the broader significance of the work and how it fits into the context of other work on these enzymes is not clear. It would be helpful to be more specific about what we know of mechanism from work on other enzymes to help the reader understand the motivation for this study.

We have added a few additional references, guided by reviewer 1 comments, that should help in this respect.